# Distinct population code for movement kinematics and changes of ongoing movements in human subthalamic nucleus

**Dennis London[1,2]\*, Arash Fazl[2], Kalman Katlowitz[2,3], Marisol Soula[2,3], Michael H Pourfar[2], Alon Y Mogilner[2†], Roozbeh Kiani[1,3,4]\*†**

[1]Center for Neural Science, New York University, New York, United States; [2]Department of Neurosurgery, Center for Neuromodulation, NYU Langone Health, New York, United States; [3]Neuroscience Institute, NYU Langone Health, New York, United States; [4]Department of Psychology, New York University, New York, United States

**Abstract** The subthalamic nucleus (STN) is theorized to globally suppress movement through connections with downstream basal ganglia structures. Current theories are supported by increased STN activity when subjects withhold an uninitiated action plan, but a critical test of these theories requires studying STN responses when an ongoing action is replaced with an alternative. We perform this test in subjects with Parkinson's disease using an extended reaching task where the movement trajectory changes mid-action. We show that STN activity decreases during action switches, contrary to prevalent theories. Furthermore, beta oscillations in the STN local field potential, which are associated with movement inhibition, do not show increased power or spiking entrainment during switches. We report an inhomogeneous population neural code in STN, with one sub-population encoding movement kinematics and direction and another encoding unexpected action switches. We suggest an elaborate neural code in STN that contributes to planning actions and changing the plans.

**\*For correspondence:**
dennis.london@nyulangone.org (DL);
roozbeh@nyu.edu (RK)

†These authors contributed equally to this work

**Competing interest:** The authors declare that no competing interests exist.

## Introduction

The ability to change an ongoing plan of action is essential to success in a dynamic world. How do we switch from executing one action to another? The gating of movements is thought to differentially engage cortico-striatal-thalamic pathways. Executing a planned movement involves activation of the direct pathway (*Kravitz et al., 2010*; *Sippy et al., 2015*), while preventing or stopping a movement involves activation of the indirect (*Kravitz et al., 2010*) and hyperdirect pathways (*Aron and Poldrack, 2006*; *Chen et al., 2020*). The subthalamic nucleus (STN), located at the intersection of the indirect and hyperdirect pathways, is theorized to globally suppress movement through excitation of the output structures of the basal ganglia (*Frank, 2006*). These theories are supported by the experimental observations that withholding a planned but uninitiated action is associated with increased firing rates in STN neurons (*Bastin et al., 2014*; *Isoda and Hikosaka, 2008*; *Pasquereau and Turner, 2017*; *Schmidt et al., 2013*) and elevated local field potential (LFP) power in the beta frequency range – a hallmark of inhibition effects of STN on action planning circuitry (*Alegre et al., 2013*; *Benis et al., 2014*; *Kuhn et al., 2004*; *Ray et al., 2012*). A prediction of the current theories is that STN contributes to switching an action plan by increasing its activity to halt the ongoing action.

However, it is possible that STN plays a more elaborate role in governing actions than just cancelling existing plans. This possibility is supported by past studies that showed STN neurons increase their activity during movement initiation (*Bastin et al., 2014*; *Pasquereau and Turner, 2017*) and

distinct LFP activity occurs during behavioral conflicts (*Zavala et al., 2013*; *Zavala et al., 2014*), suggesting that the STN may do more than globally suppress movement.

To investigate STN neural responses during action switches, we developed a novel behavioral task in which human subjects with Parkinson's disease undergoing implantation of deep brain stimulation (DBS) electrodes moved their hands on a straight or turning trajectory from a home position to target regions on either side of the home position (Reach or Planned Turn trials). On a random subset of trials, intermixed with others, subjects were cued to initiate a straight reach but then cued to switch to a turning trajectory mid-movement (Impromptu Turn trials). We recorded spiking activity of STN units and LFPs from the same sites during the task. Our new task contrasts with prior studies of STN which focused largely on the starting or withholding of simple movements, such as button presses or joystick movements in humans (*Alegre et al., 2013*; *Bastin et al., 2014*; *Benis et al., 2014*; *Ray et al., 2012*), saccades or reaches in non-human primates (*Isoda and Hikosaka, 2008*; *Pasquereau and Turner, 2017*), or nose-pokes in rats (*Schmidt et al., 2013*). Comparing STN neural responses during Impromptu Turn, Planned Turn, and Reach trials provides a framework to examine current theories on the role of STN in movement planning and execution.

We found that interruption of a planned movement with a cue to switch movement trajectory was associated with decreased firing rates in STN, contrary to predictions of prevailing theories. Careful investigation of the neural responses revealed an elaborate code, in which the population of neurons encoded the movement plan, hand kinematics during the movement, and changes in the movement plan. Unsupervised clustering of the population response dynamics revealed functional clusters of neurons, one specialized for encoding movement kinematics and the other for encoding changes in movement plan. The neurons also exhibited a wide range of response lags, some predicting upcoming changes in kinematics and some reflecting recently executed changes, creating a time reservoir for planning and tracking of actions. Finally, we show that switching the movement trajectory was associated with constant beta power in the LFP and similar entrainment of action potentials with beta frequency oscillations, further supporting the idea that switching ongoing actions has a distinct neural mechanism from withholding uninitiated actions. Together, our results suggest an elaborate population code in STN and an intricate role in planning and execution of complex actions, beyond what is predicted by existing theories.

## Results

### Diverse responses of STN units

We recorded single and multiunit spiking activity from the STN while human subjects with Parkinson's disease performed single-step or multistep reaching movements during surgeries to implant DBS electrodes. Subjects controlled a cursor on a screen with free hand movements in 3D space as recorded by a stereoscopic camera (*Figure 1A*). They used the hand contralateral to the recorded STN to move the cursor into a fixation area. After a variable-duration fixation period, they received an instruction cue to make a horizontal reaching movement to a target region (Reach trials) or a horizontal and then upward reaching movement to another target region (Planned Turn trials). Movement was allowed to begin only after a go signal (disappearance of fixation point), separating movement planning and execution phases of the trial. On a fraction of trials initially cued as Reach trials, the target changed mid-movement, instructing the subject to switch the ongoing horizontal movement to a vertical movement (Impromptu Turn trials) to reach the same target region as in the Planned Turn trials. Subjects completed all trial types with similarly high accuracies and comparable movement onset latencies (*Figure 1B,C*), with their hand trajectories following the instructed paths (*Figure 1D–F*). However, the latencies of turns in Impromptu Turn trials were systematically longer than movement onset latencies (*Figure 1C*) likely due to the overhead of switching an ongoing movement into a new movement trajectory. This latency difference suggests that the computations involved in changing an ongoing plan might be distinct from those for starting a planned movement.

We recorded from 39 units in 9 STNs of 8 subjects (one subject completed separate recording sessions from STNs of both hemispheres). Subject demographics and the number of units recorded in each subject are shown in *Supplementary file 1*. The recorded neurons showed a variety of response dynamics (*Figure 2*, *Figure 2—figure supplement 1*), suggesting an inhomogeneous neural population in the STN. To demonstrate this response diversity, we calculated the peri-stimulus time histograms

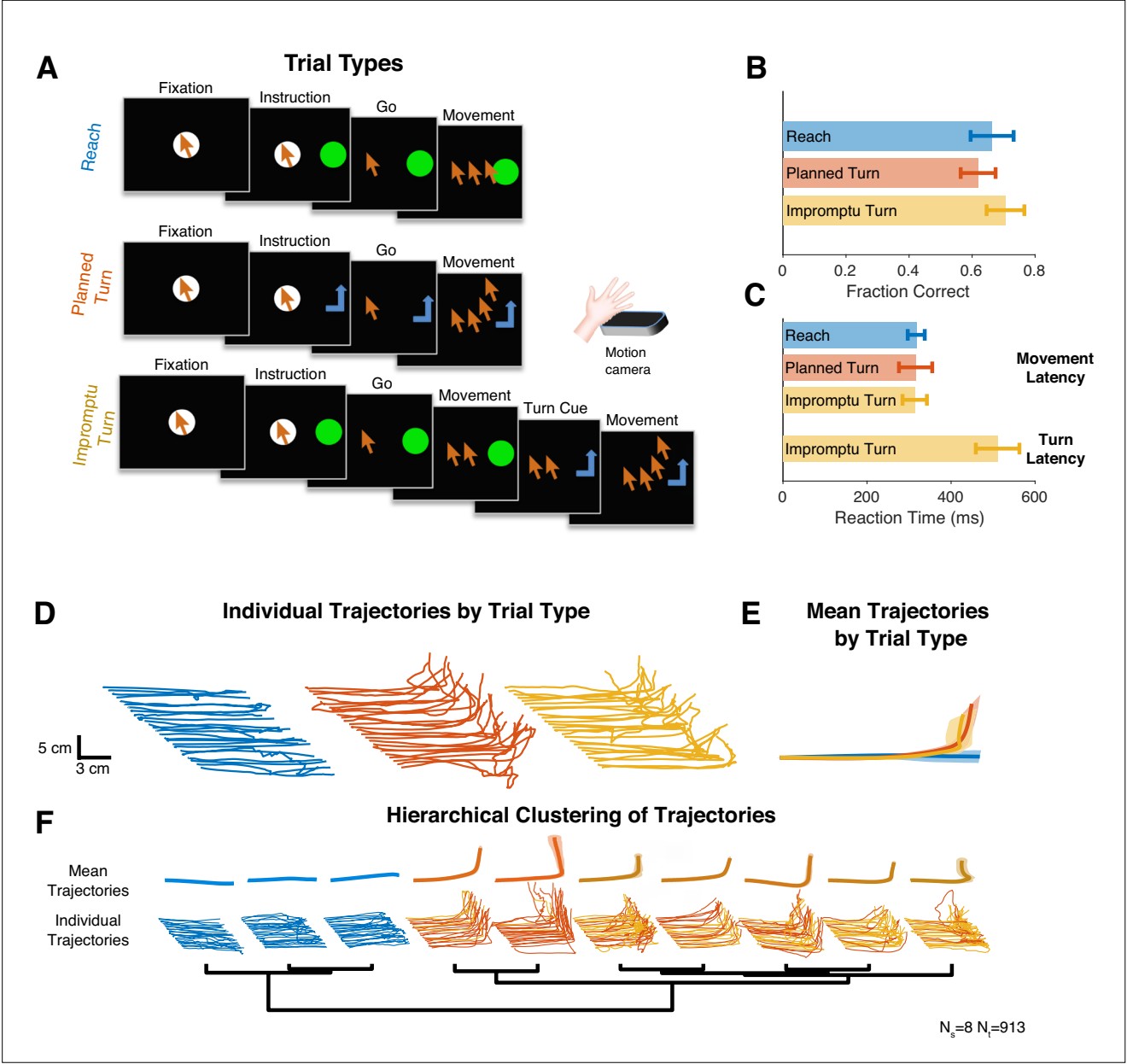

**Figure 1.** Subjects completed cued single-step and multistep reaching movements during intra-operative recordings from STN. (**A**) Subjects completed trials involving planned linear movements (Reach), planned two-step movements (Planned Turn), and trials initially cued as Reach trials with a later cue mid-movement to alter the trajectory (Impromptu Turn). Cursor sequence on the screen indicates movement trajectory. (**B, C**) Trials that met the accuracy criteria as a fraction of the total trials (**B**) and movement latency (**C**) were similar on all trial types. However, the time from the turn cue to the change in movement trajectory on Impromptu Turn trials (turn latency) was larger than the movement latency. (**D**) Example movement trajectories. Each line illustrates a sample trial. The fixation point is indicated by the most leftward point on each trajectory. Start positions are shifted horizontally and leftward trials are flipped horizontally to improve clarity and facilitate comparison of trials in the same group. (**E**) Mean movement trajectories across all subjects show successful completion of distinct movement plans across different trial types. Shading is SE. (**F**) Hierarchical clustering of movement trajectories reveals segregation of Reach trajectories from the Planned and Impromptu Turn trajectories. $N_s$ and $N_t$ indicate the number of subjects and the number of trials, respectively, across all panels.

(PSTHs) of each unit relative to the fixation, instruction, movement onset, turn onset, and feedback events for each trial type. We analyzed the population dynamics using a principal component analysis on unit PSTHs across the recorded population, focusing on the first four principal components (PCs), which explained 61 % of the variance. The population responses occupied distinct locations in this state-space at different times throughout the trial (*Figure 2E–H*), suggesting encoding of task events

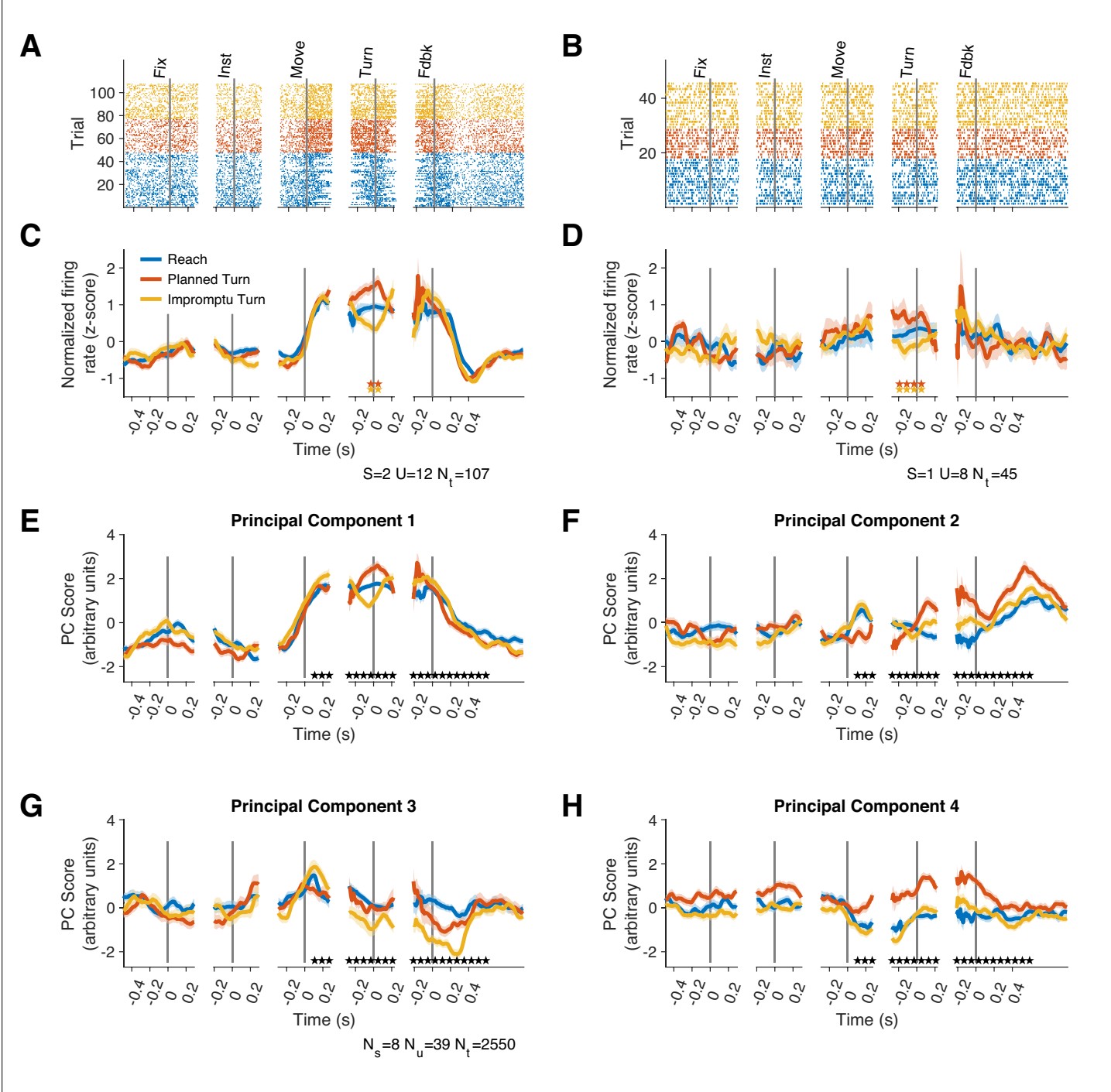

**Figure 2.** Firing rates of STN units represent different task events and movement types. (**A–D**) Raster plots (**A, B**) and corresponding PSTHs (**C, D**) of two example units. Neural responses are aligned to the fixation (Fix), instruction (Inst), movement onset (Move), turn onset (Turn), and feedback (Fdbk). Each point in (**A, B**) represents an action potential. Significant differences (p<0.1) between pairs of PSTHs within each unit are indicated by corresponding pairs of colored stars. S and U indicate subject and unit identities, respectively, and $N_t$ indicates the total number of trials per unit. (**E–H**) Projections of population responses on the top four principal components reveals distinct response patterns associated with different task events and trial types. $N_s$, $N_u$, and $N_t$ indicate the number of subjects, units, and trials, respectively. Stars indicate Holm–Bonferroni corrected p<0.001 for pairwise permutation tests. Shadings in **C–H** are SE. Because there was no turn onset on reach trials, we sampled from the distribution of turn onset times on turn trials to create the turn-aligned PSTHs for reach trials (see Materials and methods).

The online version of this article includes the following figure supplement(s) for figure 2:

**Figure supplement 1.** Sample units.

**Figure supplement 2.** Unit locations.

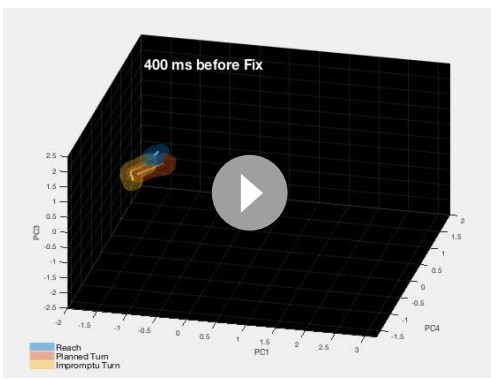

**Video 1.** State space trajectories for firing rate displayed separately for Reach, Planned Turn, and Impromptu Turn trials. Shading indicates standard error. PC2 distinguishes different trial types the least and is omitted from these plots for clarity. The first four principal components (including PC2) were included in statistical analysis of principal components data.
https://elifesciences.org/articles/64893/figures#video1

in STN. Furthermore, the population response trajectories for different trial types diverged 60 ms after movement onset and remained separated until 610 ms after the feedback (*Video 1*; Holm–Bonferroni corrected pairwise permutation tests p<0.001), indicating that the STN population distinguishes different movements both during and after their execution.

Although response patterns varied considerably across units, many units elicited similar motifs across trials. We, therefore, hypothesized that the STN neural population consists of functional clusters with a mixture of selectivities (*Figure 3A*). To better characterize the population diversity, we performed a hierarchical clustering analysis on the unit responses using the contribution of units to the first four population response PCs (see Methods). The analysis revealed two main classes that together comprised 87 % of the units (*Figure 3*): units of one group modulated their firing rates with ongoing movement kinematics (Movement Units; 41%, present in 6/8 subjects, *Figure 3B* and *Figure 2—figure supplement 1A,C,E*), and did not distinguish planned and unplanned turns, whereas the units of the other group had a more attenuated representation of movement kinematics but encoded unexpected changes of the ongoing movement plan, distinguishing different trial types (Turn Units; 46%, present in 6/8 subjects, *Figure 3C* and *Figure 2—figure supplement 1B,D,F*). Since we used unsupervised clustering of the PSTH dynamics to identify these groups, units in the same group were not perfectly homogenous and demonstrated diversity that could suggest sub-clusters within each group. To preserve our statistical power, however, we focus only on the two top clusters as a basis for deeper exploration of STN responses in the following sections.

All subjects had recordings of Movement and/or Turn Units. Our dataset includes both single and multi-units, in which most spikes originate from a single unit (*Supplementary file 1*). Single and multi-units had similar proportions in the Turn and Movement Unit groups. We did not find distinct distributions of the locations of Turn and Movement Units within STN (*Figure 2—figure supplement 2*). There was no significant difference in the median baseline firing rates of Turn and Movement Units (36.6 Hz, IQR: [22.2–56.6 Hz] vs. 35.5 Hz IQR: [15.5–88.0], respectively, p=0.82 Wilcoxon rank sum test).

## Movement Units respond to kinematic properties of movements

The firing rates of Movement Units were modulated by task events and trial types. They increased at movement onset and decreased after feedback. Firing rates were significantly lower on Planned Turn trials beginning 140 ms after movement onset and lasting until 130 ms before turn onset (*Figure 3*, p=0.035 and p=0.0070 for Planned Turn vs Reach and Planned Turn vs Impromptu Turn, respectively, Holm-Bonferroni corrected linear mixed effect models with random effects for subjects and units nested within subjects). These changes were also seen when examining individual unit PSTHs in this time interval. Despite limited trial counts for individual units, 38 % of Movement Units showed decreased firing on Planned Turn compared to Impromptu Turn trials, significantly above chance (p=5.0 × 10$^{-4}$, binomial test). However, only 19 % of Movement Units showed decreased firing on Planned Turn compared to Reach trials (p=0.21, binomial test). The difference of the firing rates at the population level afforded accurate classification of trials. A decoder trained to classify trials based on population responses achieves a cross-validated accuracy of 67% and 71% for decoding Reach from Planned Turn trials and Planned Turn from Impromptu Turn trials, respectively.

While Movement Units increased their firing rates at the onset of movement, this was not a direct response to the appearance of the go cue itself. As shown in *Figure 3—figure supplement 1A*, an increase in Movement Unit firing rate occurred ~300 ms after the go cue, a significantly longer

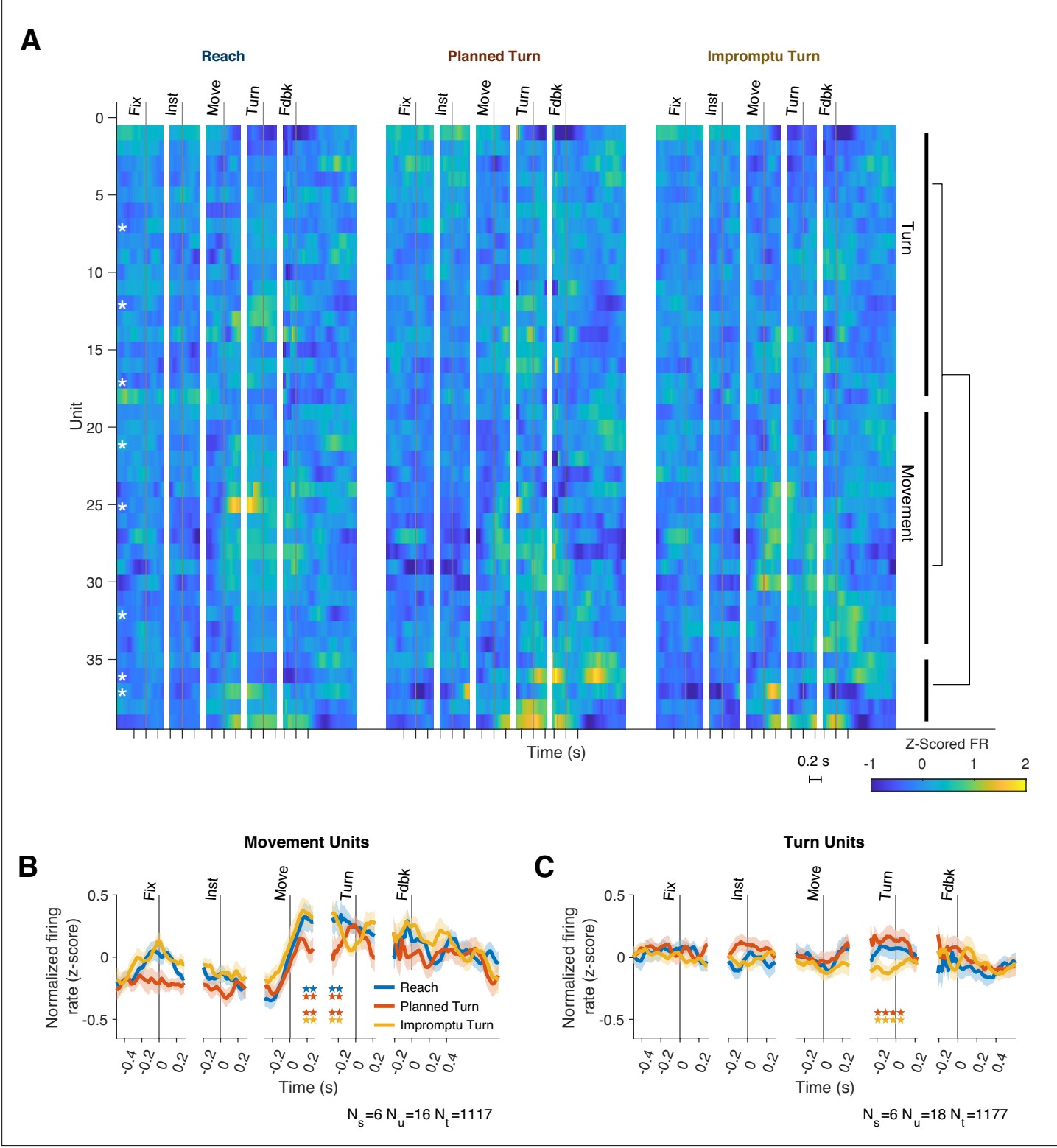

**Figure 3.** Two functional groups of neurons with distinct response dynamics. (**A**) Normalized responses of all recorded units. Hierarchical clustering of responses revealed two dominant functional groups: Movement Units (41%) and Turn Units (46%). White asterisks indicate single units, which are not clustered together. Example units from each group are shown in *Figure 2—figure supplement 1*. (**B, C**) The mean PSTHs across the two functional groups. Horizontal pairs of colored stars indicate times with significantly different firing rates for the corresponding pairs of trial types (p<0.05, Holm–Bonferroni corrected linear mixed effect models). $N_s$, $N_u$, and $N_t$ indicate the number of subjects, units, and trials represented.

*Figure 3 continued on next page*

*Figure 3 continued*

The online version of this article includes the following figure supplement(s) for figure 3:

**Figure supplement 1.** Firing rate responses aligned to cues and movements.

delay than would be expected for visual processing of the cue. In contrast, firing rates began to increase ~200 ms before the movement onset and continued to rise until after movement onset, suggesting that the increased activity is associated with movement planning and execution.

Because STN neurons are known to change their activity with the kinematics of movement (*Georgopoulos et al., 1983*; *Tankus et al., 2017*), we sought to determine if movement kinematics differed across trial types and if the pre-turn differences of firing rates could be explained by contrasting movement kinematics.

The kinematics of movement varied by trial type, as shown in *Figure 4—figure supplement 1*. During the initial fixation period, hands were stationary, but after the go signal, horizontal acceleration and horizontal speed increased on all trial types (*Figure 4—figure supplement 1A-B*). Horizontal acceleration then decreased and became negative, compatible with deceleration necessary to stop the hand at the target location. On Planned and Impromptu Turn trials, vertical acceleration and vertical speed increased at turn onset (*Figure 4—figure supplement 1C-D*). On Impromptu Turn trials, horizontal speed decreased before the vertical speed increased, and the magnitude of these changes was such that the total speed (calculated as the Euclidean norm of the vertical and horizontal speeds) decreased before turn onset. (*Figure 4—figure supplement 1E*). To facilitate pooling data across subjects and comparison of movement kinematics with neural responses, the total speed and acceleration were normalized (z-score) within each subject (*Figure 4A–D*). Mirroring the kinematic profile of subjects' hands, Movement Units increased their firing rates at movement onset on all trial types and decreased them prior to turn onset on Impromptu Turn trials (*Figure 4E*).

If the responses of Movement Units reflect *upcoming* movement kinematics, we would expect their firing rates to be a function of the speed and acceleration that would occur in the future. To test this, we modeled the firing rates of Movement Units as a function of lagged speed and acceleration (*Equation 12*). We determined the lag of each unit as the time shift that best explained the firing rate changes as a function of speed and acceleration. These kinematic models of firing rate generated predicted PSTHs that closely matched the data (inset of *Figure 4G* and *Figure 4—figure supplement 2A,C,E*; in-sample $R^2$ = 0.59, 95% CI: [0.57 0.61], out-of-sample $R^2$ = 0.48, 95% CI: [0.45 0.51] across units) and explained the firing rate dynamics (speed mean $\beta$ = 0.19, 95% CI: [0.01 0.38]; acceleration mean $\beta$ = 0.25, 95% CI: [0.07 0.43]). However, the best-fitting models had lags that varied considerably in the population of Movement Units (*Figure 4G* and *Figure 4—figure supplement 2G*), ranging from +190 ms to -110 ms. Therefore, some units predicted future movement kinematics (positive lags; 56 % of units, mean lag± s.e., 110 ms ± 24 ms), whereas others tracked past movement kinematics (negative lags; 44 % of units, mean lag± s.e., –89 ms ± 13 ms). Overall, the population of Movement Units represented the kinematics of future movements as well as those of past and ongoing movements.

For the results above, we used a model that explained neural responses based on total movement speed and acceleration, regardless of movement direction. To test whether acceleration and speed along particular movement directions were more effective at modulating neural responses, we explored a multitude of alternative models in which the firing rate was a function of acceleration and speed along the horizontal and/or vertical directions on the screen. These models were fit separately for each unit and the model with the highest $R^2$ for the held-out data (out-of-sample $R^2$) across all units was chosen (see Methods for description of alternative models and process for holding out data). The best model was identical to the main model in the paragraphs above and contained two predictors: the total speed and total acceleration. The alternative models underperformed in predicting the held-out data. But among them, models that included total speed or total acceleration tended to explain response modulations better than the models that included only directional speed and acceleration (*Figure 4—figure supplement 3*). Overall, our results suggest that kinematic-dependent response modulations of Movement Units are not strongly dependent on the direction of movement.

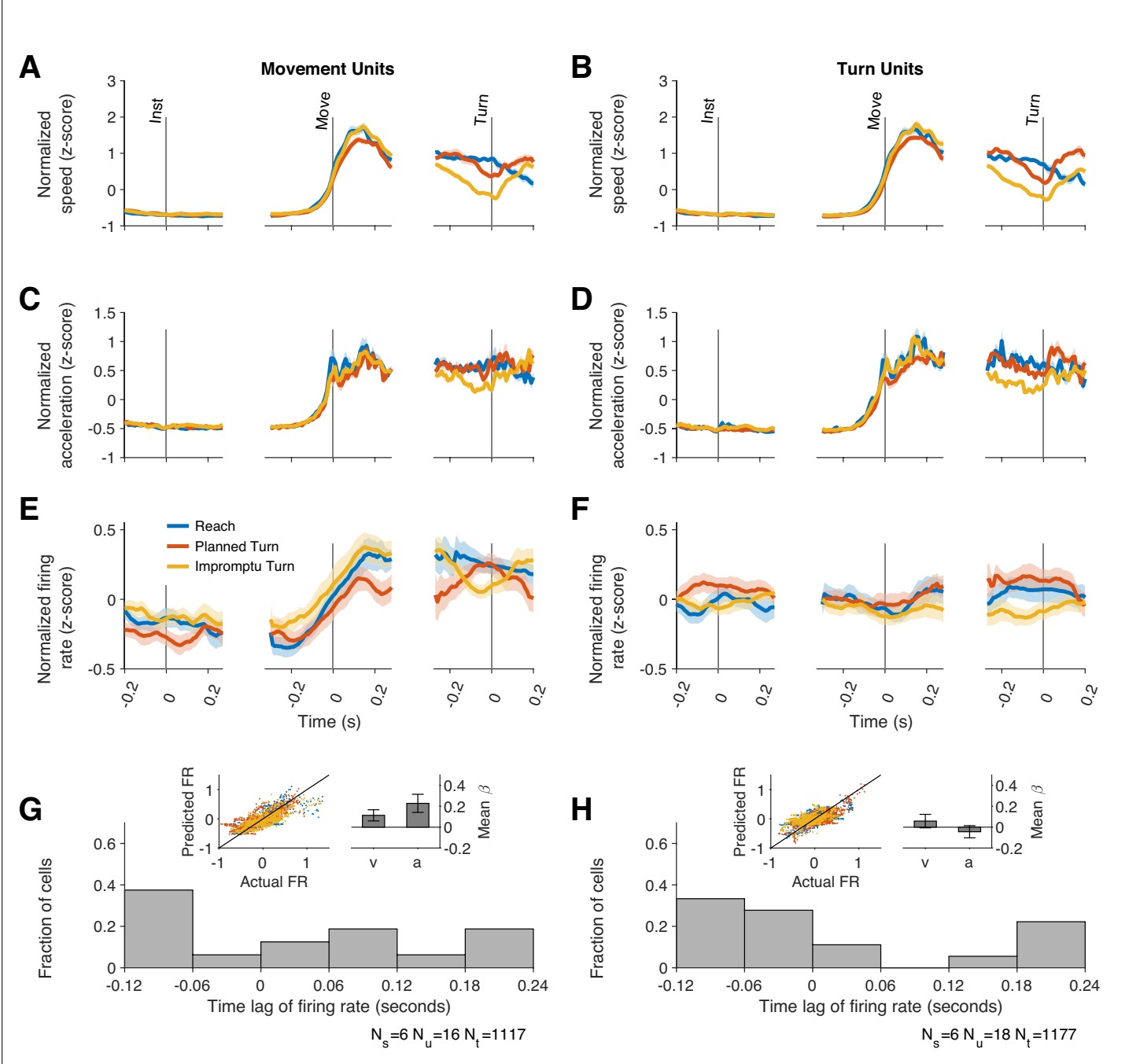

**Figure 4.** Firing rates of Movement Units reflect kinematic parameters of past, ongoing, and future movements. (**A, B**) Average movement speed during recordings from Movement and Turn Units. Movement speed was z-scored across trials of each session before averaging across sessions. (**C, D**) Average movement acceleration. (**E–F**) Average firing rates of Movement and Turn Units aligned to task events. These panels are identical to those in *Figure 3B–C* and replicated here to facilitate comparison of firing rate dynamics with changes of movement speed and acceleration. Shading indicates SE in **A–F**. (**G–H**) The firing rate of each unit was modeled as a linear function of time-lagged speed and acceleration (*Equation 12*). The distributions of best-fitting time lags are shown across units. Positive lags indicate that firing rates predict future movement kinematics, and negative lags indicate that firing rates reflect past movement kinematics. Insets display the actual and predicted firing rates and the regression coefficients for speed and acceleration. Error bars are 95 % confidence intervals.

The online version of this article includes the following figure supplement(s) for figure 4:

**Figure supplement 1.** Movement speed and acceleration.

**Figure supplement 2.** Kinematic model of firing rate responses.

*Figure 4 continued on next page*

*Figure 4 continued*

**Figure supplement 3.** Alternative models and model selection for Movement Units.

**Figure supplement 4.** Alternative models and model selection for Turn Units.

## Movement Units increase firing rate for medial movements

The models above demonstrate that the modulation of Movement Unit responses with movement speed and acceleration is not directionally tuned. However, that does not necessarily imply that these units lack other forms of directional tuning. In particular, Movement Units have higher mean firing rates for the medial movement direction but similar kinematic-dependent response modulations around the mean for different movement directions. In our task, the instructed movement direction was either medial (towards the recorded STN) or lateral (away from the recorded STN) to the recorded STN. For successful completion of trials, subjects had to select not only the correct movement type (Reach vs. Turn) but also the correct direction (medial vs. lateral). The mean firing rates across trial types from 10 ms before to 270 ms after movement onset were higher on trials where the movement direction was medial, as shown in *Figure 5A–D,H* (p=0.019 linear mixed effect model). This effect was seen in 8/12 individually examined Movement Units (p=$1.7 \times 10^{-7}$, binomial test), and a population response decoder trained on population responses can separate medial from lateral trials with an accuracy of 80 %.

Since the firing rate of these units is also associated with movement kinematics, we tested whether the mean firing rate difference could be ascribed to different speeds and kinematics on medial compared to lateral movements. However, the speed and acceleration were comparable on medial and lateral trials (*Figure 5—figure supplement 1*). Furthermore, we used the kinematic model of each unit to predict its firing rate for medial and lateral movements. Residual firing rates around these predictions offer a test for the hypothesis that kinematic differences account for the mean firing rate differences on medial and lateral trials. This hypothesis would predict no systematic discrepancy in the residuals as the model would have already captured the effects of kinematic differences on firing rates. Contrary to this prediction, the model residuals reflected systematic firing rate differences, similar to those explained in the previous paragraph (*Figure 5E–H*) with the mean residual firing rate across this interval greater for medial trials than lateral trials (p=0.034, linear mixed effect model). A population response decoder is still able to separate trial types based on residual firing rates; its accuracy is 70 %. Therefore, Movement Units showed a sizeable difference in firing rates around movement onset of medial and lateral trials. However, their firing rate modulations with movement kinematics were not noticeably directional.

If movement direction affected firing rate, why did models that incorporated the x- and y- components of speed and acceleration perform poorly predicting the firing rate? Those models could accommodate differential modulation of firing rates with the kinematics of medial and lateral movements. For example, if firing rates changed by different amounts with changes of speed on medial compared to lateral movements, models with directional components would outperform. This did not occur in the data; firing rate modulated positively and by the same amounts with the magnitude of speed and acceleration on both medial and lateral trials, but the mean firing rate was overall lower on lateral trials. These differences between medial and lateral trials were only present after movement onset and disappeared prior to Turn Onset. Thus, there appears to be a firing rate offset determined by movement direction that occurs after movement onset.

## Turn Units predict whether an ongoing movement plan will change

Movement Unit PSTHs exhibited their greatest modulation at movement onset and at feedback (*Figure 3B*), which coincided with movement cessation. In contrast, Turn Units changed their firing rates prior to turn onset, as shown in *Figure 3C*. Between 240 ms before to 80 ms after turn onset, the firing rate on Impromptu Turn trials was significantly smaller than on Planned Turn trials (p=0.0031, Holm–Bonferroni corrected linear mixed effect model). Despite limited trial data, 33 % (6/18) of individual Turn Units had significantly smaller firing rates on Impromptu Turn than Planned Turn trials in this interval, significantly more than expected by chance (p=0.0012, binomial test). The difference in firing rates between Planned and Impromptu Turns is 0.2 z-scores, which represents half of the range of modulation of firing rates of Turn Units prior to feedback. This difference is sufficient to decode

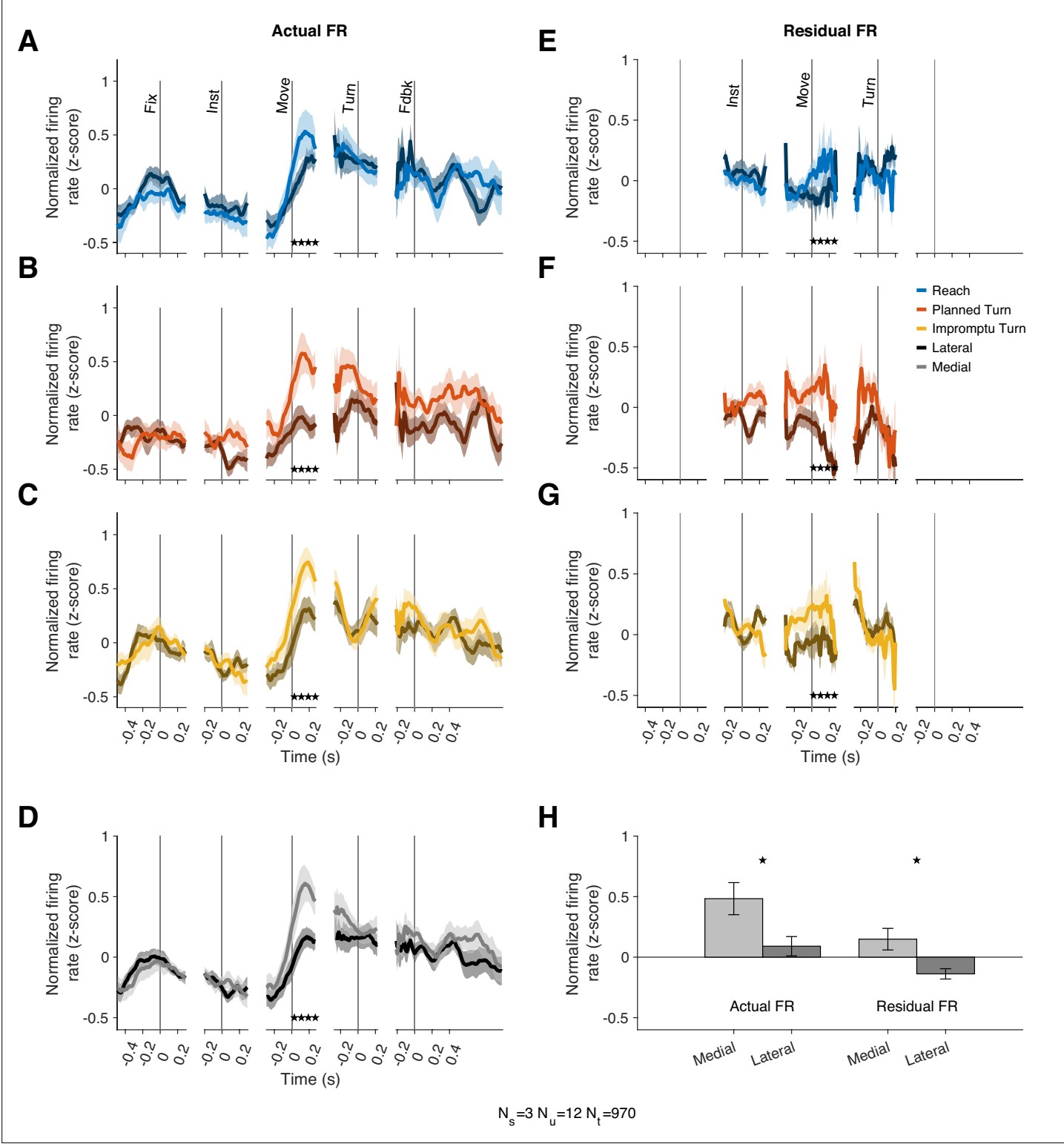

**Figure 5.** Movement Units have higher firing rates at the onset of medial movements. (**A–C**) PSTHs separated by trial type and medial (lighter shades) vs lateral movements (darker shades). Stars denote times with statistically significant differences between medial and lateral movements. (**D**) Average PSTHs for medial and lateral movements across all trial types. (**E–G**) Residual firing rate PSTHs for different trial types after regressing out kinematic effects. (**H**) Average firing rates in a window starting 10 ms before and ending 270 ms after movement onset. Stars denote p<0.05 (linear mixed effect model). Error bars in all panels are SE.

The online version of this article includes the following figure supplement(s) for figure 5:

*Figure 5 continued on next page*

*Figure 5 continued*

**Figure supplement 1.** Kinematics of medial and lateral trials.

**Figure supplement 2.** Turn units do not encode movement direction.

Planned and Impromptu Turn trials from firing rates; a decoder trained to separate Planned Turn and Impromptu Turn trials based on population responses achieves an accuracy of 74 %.

Can this difference be explained by movement kinematics? Like Movement Units, Turn Unit firing rates were also correlated with the lagged kinematics (*Figure 4H*). The best-fit lags were similarly widely distributed with firing rate lags ranging between 190 ms before to 110 ms after changes in movement kinematics (*Figure 4H* and *Figure 4—figure supplement 2H*). However, the strength of this relationship was considerably weaker than for Movement Units (out-of-sample $R^2$, Turn Units 0.13, 95% CI: [0.085 0.18], Movements Units 0.48, 95% CI: [0.45 0.51]; in-sample $R^2$, Turn Units, 0.47, 95% CI: [0.44 0.49], Movement Units 0.59, 95% CI [0.57 0.61]). Furthermore, movement kinematics did not accurately predict the PSTHs of these units (*Figure 4—figure supplement 2B,D,F*). As with Movement Units, models that included separate predictors for the horizontal and vertical components of velocity and acceleration underperformed as shown in *Figure 4—figure supplement 4*. We used the residual firing rates to determine whether movement kinematics explained away the modulation of firing rates prior to Turn Onset. As in previous sections, residual firing rates close to zero would imply that firing rate modulations before turn onset were explained by the changes in speed and acceleration. In contrast, the mean residual firing rates of Turn Units across the same interval (240 ms before to 80 ms after Turn Onset) were significantly greater than zero on Planned Turn trials (p=0.028, Holm–Bonferroni-corrected Wilcoxon-signed rank test). Furthermore, the mean residual firing rate over this interval was smaller on Impromptu Turn trials than on Planned Turn trials (p=0.0069 and Holm–Bonferroni corrected linear mixed effect model). Similar results were obtained with a variety of kinematic models. For example, an alternate model with total acceleration as the only predictor, which provided slightly better out-of-sample $R^2$ for the Turn Units alone (*Figure 4—figure supplement 4C*), also had smaller residual firing rates on Impromptu Turn trials than Planned Turn trials over this interval (p=0.0031, Holm–Bonferroni corrected linear mixed effect model). In fact, using the residual firing rates does not appreciably change the decoding accuracy (72% vs 74 % for the full responses), suggesting that movement kinematics minimally contributed to the decoder's success. Therefore, independent of changes in movement kinematics, Turn Unit firing rates were significantly smaller before turn onset on Impromptu Turn trials compared to Planned Turn trials. Unlike Movement Units, Turn Units did not display any difference when comparing medial and lateral trials (*Figure 5—figure supplement 2*), and they did not respond to the go or turn cues (*Figure 3—figure supplement 1B*). Overall, Turn Units provide a clear code for changes in ongoing action plans.

## Population activity in STN encoding movement direction and movement type precedes the movement

The response properties of the Movement and Turn Units suggest that the population of STN neurons simultaneously encode the kinematics of movements, as well as movement directions and changes of ongoing motor plans. To gain insight into the dynamics of the representation of movement directions and plan changes, we used our kinematic models to remove response fluctuations related to movement speed and acceleration. We then performed a principal components analysis on the residual firing rates of all units to identify dominant response modulations unrelated to movement kinematics. This allowed us to capture population dynamics hidden in the analyses of mean residual firing rates (*Figure 5* and *Figure 5—figure supplement 2*).

*Figure 6A–I* shows distinct population activity for medial and lateral trials from 30 ms after the instruction to 80 ms after movement onset. The mean squared distance between trajectories over this time period was higher than expected by chance (*Figure 6K*, p<0.001, permutation test; also see *Video 2* for state space trajectories). Thus, medial and lateral trials were associated with distinct population responses well before movement onset and for tens of milliseconds following the movement onset.

The population activity also distinguished different movement types prior to movement onset. Planned Turn trials showed distinct activity patterns from Reach trials between 130 ms before until 60 ms after movement onset (*Figure 6J*; p=0.024, Holm–Bonferroni corrected permutation test).

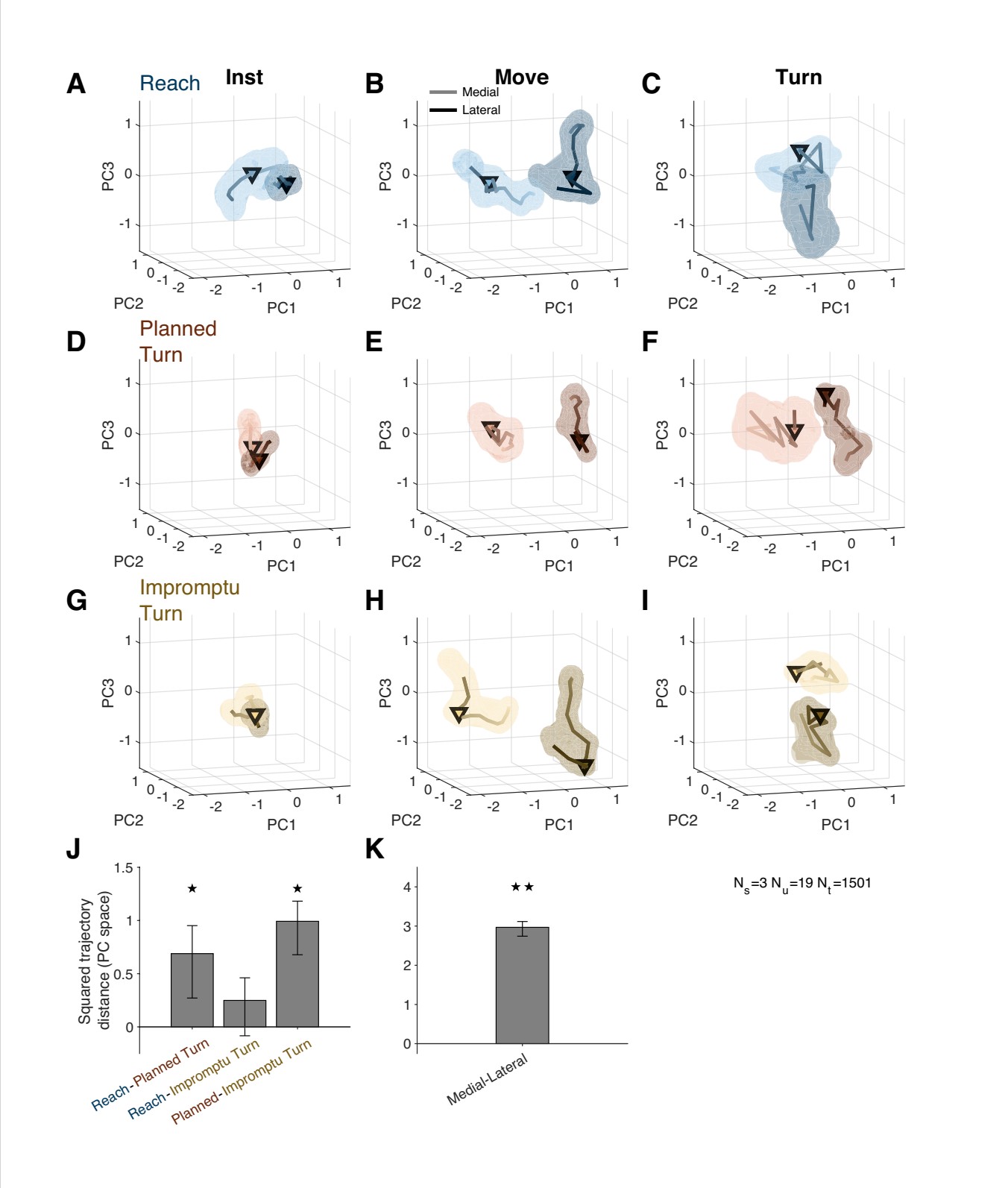

**Figure 6.** Distinct population response dynamics for different movement directions and trial types. (**A–I**) Residual firing rate trajectories in 3D principal component space for different trial types and around different task events. Later times are indicated by darker colors. Pairs of trajectories in each panel correspond to medial and lateral trials. Triangles indicate alignment times: instruction (first column), movement onset (second column), and turn onset (third column). Shading indicates standard error. (**J**) Mean squared distance of trajectories of pairs of trial types around movement onset relative to that

*Figure 6 continued*

expected by chance. A distance equal to zero indicates that the difference is not distinct from chance. (**K**) Mean squared distance between medial and lateral trajectories around movement onset relative to that expected by chance. The distance is aggregated across all trial types. ★ and ★★ indicate p<0.05 and p<0.001, respectively (permutation test). Error bars are 95 % confidence intervals.

A similar difference was also present between Planned Turn and Impromptu Turn trials (*Figure 6J*, p=0.002, Holm–Bonferroni corrected permutation test). There was no difference between activity patterns on Reach and Impromptu Turn trials (p=0.087, Holm–Bonferroni corrected permutation test). This is expected since Impromptu Turn trials begin as Reach trials.

## Movement and turn units desynchronize from beta band oscillations at movement onset

Neuronal spiking in STN is typically phase-locked to beta oscillations (*Kühn et al., 2005*; *Weinberger et al., 2006*), which are themselves shaped by inhibition in the local circuit and related to movement suppression. These response properties are hypothesized to be key to the role of STN for inhibitory control of movement in the cortical-basal ganglia network. Since changes in firing rate of the two subpopulations of STN neurons in our study were associated with changes in movement, we examined how these changes related to spike-field locking. Existing theories predict that locking of action potentials to beta oscillations would decrease prior to the movement onset. Furthermore, they predict that beta spike-field locking might increase prior to the turn onset on Impromptu Turn trials, when subjects decrease their speed and halt horizontal movement before initiating a vertical movement toward the new target location (*Figure 4—figure supplement 1*).

Both Movement and Turn Units spiked in synchrony with beta oscillations as quantified by the Pairwise-Phase Consistency (PPC), shown in *Figure 7*. PPC averaged across all trial types and times was significantly different from zero for Movement and Turn Units (median PPC, 0.0042, IQR: [0.0014, 0.010], p=0.0023 for Movement Units, and 0.0056, IQR: [$6.6 \times 10^{-4}$, 0.026], p=0.0033 for Turn Units, Wilcoxon signed-rank tests). This corresponded to a firing rate peak-to-trough modulation of 30% and 35%, respectively, for Movement and Turn Units.

However, contrary to past theoretical predictions, PPC was not significantly different between trial types (no significant clusters detected by cluster mass test, PPC averaged across movement and turn epochs not different between trial types, p=0.94, p=0.68, p=0.93, for Movement Units, Turn Units, and the combined neural population, respectively, Friedman's test). Furthermore, the action potentials of Movement Units were uncoupled from the beta oscillation around the time of the turn onset (median PPC = $5.1 \times 10^{-4}$, IQR: [−0.0045, 0.0043], Wilcoxon signed-rank test p=0.88). Notably, the uncoupling occurred even earlier for Turn Units, around the time of movement onset (median PPC = 0.0020 IQR: [$−8.6 \times 10^{-4}$, 0.012], p=0.064, Wilcoxon signed-rank test; compare *Figure 7I,J*). Compatible with these observations, the induced beta power did not differ across trial types (no significant clusters detected by cluster mass test, induced power averaged across movement and turn epochs not different between trial types, p=0.72, Friedman's test), decreased below baseline shortly before movement onset, and returned to baseline after feedback (*Figure 7—figure supplement 1*). These negative results were not simple byproducts of lack of statistical power. Based on our sample size and measurement noise, our Friedman's test could detect median PPC differences across trial types as small as 0.0054 for Movement Units, $8.9 \times 10^{-4}$ for Turn Units, and $9.2 \times 10^{-4}$ for the combined neural population (significance p-value threshold set to 0.05 for the power analyses). Also, our signed rank test could detect median action potential couplings with beta oscillations as small as 0.004 for Movement Units and 0.003 for Turn Units, well within the range observed in other task epochs (*Figure 7G–J*). Furthermore, the detection threshold for the beta power change was 0.056, again well within the plausible changes in experimental observations. Overall, there was an absence of noticeable increase in beta power and the phase locking of action potentials to beta oscillations prior to turn onset on Impromptu Turn trials. In addition, firing rates increased during the

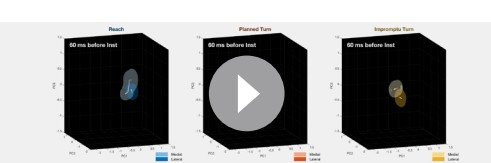

**Video 2.** State space trajectories for residual firing rate after removing the contribution of kinematics separated by trial type and direction (medial vs lateral).
https://elifesciences.org/articles/64893/figures#video2

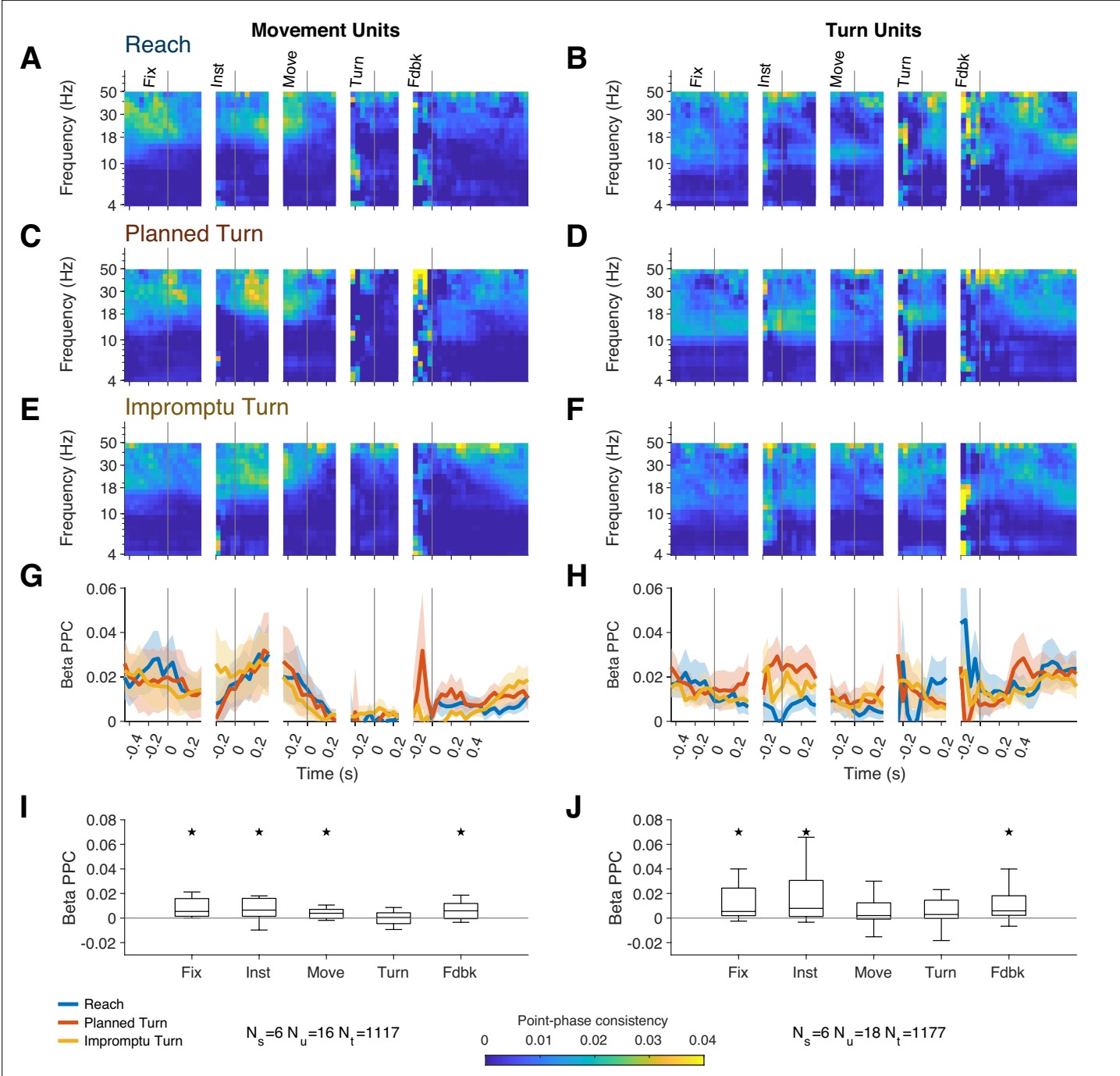

**Figure 7.** STN units fire in synchrony with beta oscillations except during the movement phase of the task. (**A–F**) Pairwise-phase consistency as a function of time and frequency for Reach (**A, B**), Planned Turn (**C, D**), and Impromptu Turn (**E, F**) trials shows decreased synchronization between spikes and LFP in the beta range around movement onset and turn onset. (**G–H**) Changes of PPC within the beta band aligned to task events. Shading indicates SE. There was no significant difference between trial types. (**I, J**) Box plots show the median and interquartile range of the beta band PPC across all trial types in the analysis windows for each task event. Whiskers indicate the range of the data excluding outliers, which are defined as more extreme than 1.5× IQR outside of the IQR (stars indicate PPC significantly different from 0, p<0.05, Holm–Bonferroni corrected Wilcoxon signed-rank test).

The online version of this article includes the following figure supplement(s) for figure 7:

**Figure supplement 1.** LFP responses.

movement phases of the trials (*Figures 2 and 3*, and *Figure 2—figure supplement 1*), while beta power and beta spike-phase locking decreased, suggesting that spiking of STN neurons switches from beta entrainment to alternative patterns during the movement phases of the trial including around turn onset. These observations further suggest that the underlying mechanisms for changes in ongoing action plans are different from those involved in withholding an uninitiated plan (e.g., stop signal tasks), corroborating our earlier conclusion based on the reduction of firing rates prior to turn onset on Impromptu Turn trials.

## Discussion

The subthalamic nucleus' location at the junction of the indirect and hyperdirect basal ganglia pathways and its synchronous activity in Parkinson's disease have led to the hypothesis that its main role is to inhibit movement (*Frank, 2006*). Here we use a novel temporally extended reaching task to show that neuronal activity in human STN increases with increasing movement speed and acceleration and decreases prior to changes in an ongoing movement plan. Changes of movement kinematics and motor plans are encoded distinctly in the STN population and represented by different subpopulations within STN. The firing rates of Movement Units increase linearly with speed and acceleration, and while they encode the kinematic changes associated with a change of plan, they do not represent the change of plan itself. In contrast, the firing rates of Turn Units primarily respond to an unpredictable need to switch movement trajectory independent of ongoing movement kinematics.

Our study is the first in humans to examine the differential role of STN neuronal activity during planned and unplanned changes of movements. Our task had three trial types, Reach, Planned Turn, and Impromptu Turn; the latter two consisted of similar two-step trajectories but were distinguished by the presence of an unpredictable cue on Impromptu Turn trials that instructed subjects to switch their ongoing reaching movement to a turning movement. We found that beginning 240 ms before the switch in movement trajectory, population activity was significantly decreased on Impromptu Turn trials compared to Planned Turn trials. Furthermore, we found no changes in induced beta power or phase locking of spike times to beta oscillations during this period.

Our results contrast with STN activity during suppression of uninitiated movement plans, which have commonly been associated with increased spiking activity in the STN population (*Bastin et al., 2014*; *Isoda and Hikosaka, 2008*; *Pasquereau and Turner, 2017*; *Schmidt et al., 2013*) and increased beta band activity (*Bastin et al., 2014*; *Ray et al., 2012*). Prior studies in STN focused on Go/No-Go (*Isoda and Hikosaka, 2008*; *Pasquereau and Turner, 2017*) or Stop Signal tasks (*Bastin et al., 2014*; *Pasquereau and Turner, 2017*; *Ray et al., 2012*; *Schmidt et al., 2013*). In Go/No-Go tasks, subjects choose from one of two actions: movement or withholding of movement. Likewise, in Stop Signal tasks, subjects are cued to make a movement and subsequently cued to withhold it after a short gap from the initial cue. In both cases, successful No-Go or Stop trials involve no movement at all. In contrast, on Impromptu Turn trials in our task, subjects are already executing a movement when they are cued to change it; they then convert their ongoing movement into a new one and change trajectory. We observed a decrease in the firing rate of Turn Units independent of ongoing kinematics in contrast to the increase in firing rate on No-Go Units observed by *Isoda and Hikosaka, 2008* and *Pasquereau and Turner, 2017*. In addition, we did not see an increase in beta power or beta range spike-phase locking around the unplanned turns though beta power was overall decreased after movement onset. It is possible that the decrease in activity we observed prior to changes in ongoing actions is reflective of the STN's role in interrupting action sequences. However, this would not be explained by existing theories of STN function, which predict an increase in activity. The decreased firing rates may reflect an alternative mechanism or additional role for STN in action control.

In our task, Turn Unit firing rates were highest prior to the turn on trials where subjects had planned to make this movement. They were lowest prior to the turn when subjects had to change their ongoing movement. And they were between these extremes on trials where subjects executed a planned movement but knew that they could be cued to switch movements. An intriguing possibility is that Turn Units may reflect the moment-by-moment confidence or expectation to complete an ongoing or recently planned movement. Consistent with this, at movement onset, the firing rates of Turn Units were at their baseline and not distinct between trials; at this point in the trial, subjects have to make the same movement regardless of cues that may occur in the future. They should thus have the same level of confidence in their initial movement.

While it is possible that our subjects' underlying Parkinson's disease influenced our data, this is not sufficient to explain our novel results. Though the STN in Parkinson's disease contains hypersynchronous beta activity (*Brittain and Brown, 2014*), beta activity increases during stopping in both Parkinson's disease (*Ray et al., 2012*) and obsessive compulsive disorder (OCD) (*Bastin et al., 2014*). Furthermore, the involvement of beta oscillations and spiking activity in response inhibition has been consistent in patients with Parkinson's disease (*Alegre et al., 2013*; *Benis et al., 2014*; *Ray et al., 2012*; *Zaghloul et al., 2012*) and OCD (*Bastin et al., 2014*) as well as in healthy non-human primates (*Isoda and Hikosaka, 2008*; *Pasquereau and Turner, 2017*) and rodents (*Schmidt et al., 2013*). Our observation that beta activity and beta-locked spiking decrease during movement is also consistent with this framework. However, beta activity and beta-locked spiking did not increase with unplanned movements, suggesting that different mechanisms underlie altering ongoing actions and preventing uninitiated ones. Future studies will determine if this finding can be replicated in non-Parkinsonian subjects.

While Turn Units may encode subjects' confidence about a selected action, Movement Units seem to more directly encode the kinematics of movement, consistent with studies going back several decades (*Georgopoulos et al., 1983*). Recently, *Tankus et al., 2017* showed that the correlation of firing rates and movement kinematics of uncued movements is lagged by different time periods. We replicated this finding in cued, task-based movements, showing that the firing rate of Movement Units predicts future speed and acceleration or reflects past speed and acceleration. By examining alternative firing rate models, we show that the modulation of Movement Unit activity with movement kinematics is not directionally tuned and is best modeled as functions of total speed and acceleration, quantified as the vector norm of speed or acceleration in the movement plane. In contrast, movement direction is represented in the overall activity of the units and seems to be distinctly coded from speed and acceleration (*Figures 5 and 6*). Thus, the neural codes for movement direction and kinematics are linearly separable at each moment.

The varying time lags of STN unit responses relative to kinematics could indicate a functional division, in which some neurons are involved in prospective planning of movements, whereas others monitor past movements, implementing a time reservoir of kinematic information for recent movements. The neurons that represent a recent movement could reflect past responses of the STN neurons that lead movements, represent proprioceptive inputs to STN (*Abosch et al., 2002*; *Rodriguez-Oroz et al., 2001*), or reflect inputs from the motor cortices that project to STN, either directly (*Aron and Poldrack, 2006*; *Chen et al., 2020*) or indirectly through other nodes of the basal ganglia. While proprioceptive feedback cannot shape responses of STN units that encode future movements, cortical inputs could contribute to the activity of those neurons. The same cortical projections could also play a role in shaping the responses of Turn Units.

Prior studies have found differences in neural activity during preparation of movements in different directions (*Isoda and Hikosaka, 2008*; *Zaghloul et al., 2012*). Our results complement past studies by showing a similar difference during the movement itself. Examining population activity after regressing out the kinematic effects, we found that medial and lateral movements are associated with distinct population responses, starting 30 ms after the instruction (at least 190 ms before movement onset) and lasting until 80 ms after movement onset (*Figure 6*). Within the STN population, the mean residual activity of Movement Units after removal of kinematic modulations distinguished medial and lateral movement directions until 270 ms after Movement Onset (*Figure 5*). In addition to the movement direction, the population activity of different sub-populations of STN neurons distinguished different trial types around key task events (*Figures 3 and 6*). Thus the population activity in STN encodes diverse properties of the movement plan and is not limited to encoding only the kinematics.

The division of the STN population into two major sub-populations – Movement Units and Turn Units – through unsupervised clustering indicates substantial differences in response properties of the two sub-populations. However, the two sub-populations also share some response properties. For example, Turn Units are moderately responsive to movement kinematics, and the residual firing rates of Movement Units show a trend toward reduction before unplanned turns. Further, the diversity of responses within each sub-population suggests the possibility that each may comprise multiple sub-groups. These sub-groups could be better characterized in a larger dataset, and especially by recording from the same neurons in diverse tasks and movements. Because of partial overlap between response properties of Movement and Turn Units, it is also reasonable to treat the STN population as

a mixture of neurons with diverse selectivity. In fact, adopting such a perspective and analyzing the STN population responses was quite effective at revealing that the movement plans for different trial types as well as movement directions are encoded hundreds of milliseconds before movement onset. However, mechanistic understanding of the population response requires accurate characterization of individual neurons in the population and discovering distinct sub-populations within the circuit.

An intriguing hypothesis is that Movement Units encode kinematic information to contribute to execution and possibly planning of actions, whereas Turn Units contribute to maintaining or changing ongoing action plans. Since Movement Unit activity is predictive of future kinematics, this information is likely rapidly transmitted directly from cortex via the glutamatergic hyperdirect pathway. STN neurons that receive these glutamatergic inputs project to GABAergic SNr neurons (*Xiao et al., 2015*). Compatible with these anatomical connections, stop signals emerge in the STN neural responses earlier than in substantia nigra pars reticulata (SNr) responses (*Schmidt et al., 2013*). Furthermore, stopping has been linked to activity in the projections from inferior frontal gyrus to STN (*Chen et al., 2020*; *Swann et al., 2009*) and low-frequency stimulation of glutamatergic afferents in STN (i.e., the hyperdirect pathway) worsens bradykinesia though stimulation of the motor cortex-STN projection improves it (*Gradinaru et al., 2009*). We suggest that Movement Units in STN may exert their influence on movement control through this inhibitory pathway. Because the representation of movement kinematics is much weaker in the activity of Turn Units, they likely have different inputs compared to Movement Units, possibly including inputs from the indirect pathway. These insights offer testable hypotheses for future studies on the neural mechanisms that underlie changing ongoing plans of action.

Our results support a versatile and diverse neural code in STN capable of supporting a variety of functions in planning, executing, monitoring, and changing plans of action. As *Pasquereau and Turner, 2017* note, this richness in neural code appears incompatible with a narrowly defined role for STN that includes only movement inhibition. We therefore suggest a broader role for STN in motor control. Future studies should explore the rich neural code in STN and clarify its causal role in various aspects of motor control.

## Materials and methods
### Subjects
Subjects were patients with Parkinson's disease undergoing implantation of deep brain stimulation electrodes in the subthalamic nucleus (see *Supplementary file 1* for demographic information on subjects). They were recruited at their regular appointments, signed informed consent for participation in the study, and completed a training session on the behavioral task at this time. They received monetary compensation for participation in the training session and intra-operative session. All experimental procedures were approved by the Institutional Review Board at NYU Langone Medical Center.

### Surgery and electrophysiological recordings
Subjects did not take their anti-Parkinsonian medications on the day of surgery. Surgical implantation was staged; the DBS lead in each hemisphere and the pulse generator were implanted in separate surgeries. Subjects were given moderate sedation for application of the Leksell stereotactic frame (Elekta, Stockholm, Sweden) and underwent a CT scan for frame localization, which was then aligned with a preoperative structural MRI with gadolinium.

Microelectrodes were targeted to the ventral border of the STN based on structural MRI and the AC-PC coordinates (11 mm lateral to midline, 5 mm inferior to the AC-PC plane, and 4 mm posterior to the AC-PC midpoint). The trajectory to the target was chosen such that the skull entry point was in the vicinity of the coronal suture and modified to avoid structures including vessels, ventricles, and the caudate nucleus. Sedation was typically discontinued after making the incision, and subjects were awake on minimal or no sedation for the electrophysiological recordings. Recordings were performed from a single microelectrode or concurrently from two microelectrodes spaced 2 mm apart at their insertion. For each electrode, an outer cannula was advanced to 15 mm above the target. A microelectrode in an inner cannula (Neuroprobe Sonus Shielded tungsten microelectrode, 1 MΩ impedance, Alpha Omega, Alpharetta, GA) was advanced to the end of the outer cannula and further advanced with a microdrive (Alpha Omega, Alpharetta, GA). When two microelectrodes were used,

they were driven concurrently on the same microdrive. The microelectrode tip protruded 3 mm distal to the end of the inner cannula. Recordings were referenced to the outer cannula. Online analysis of spiking activity was performed with the Alpha Omega recording system.

The recorded units spanned the space between the dorsal and ventral borders of STN. Entrance to the STN was noted based on two criteria: (1) real-time alignment of a CT scan performed at the time of surgery with a high-resolution pre-surgical structural MRI, and (2) an increase in background spiking (*Novak et al., 2007*) and identification of rhythmically firing cells at high firing rates (*Hutchison et al., 1998*). A decrease in background spiking marked the ventral boundary of STN. In many of our surgeries deeper cells along this trajectory fired rapidly with a relatively constant firing rate, characteristic of SNr cells. This procedure has previously been described in detail by our group (*Sterio et al., 2002*).

We used the pre-operative high resolution structural MRI with gadolinium, pre-operative high resolution CT scan with the Leksell frame in place, planned trajectory, and intra-operative or post-operative CT scan with DBS lead in place to localize our recordings. Our procedure is as follows. The pre-operatively planned target in AC-PC coordinates was identified on MRI. This MRI was merged with the pre-operative frame-localizing CT scan. Using the frame localization, the target was converted into frame coordinates. The pre-operatively planned trajectory was identified using this target and the ring and arc angle of the Leksell frame. Based on the alignment of the post-operative CT scan, which shows the DBS lead, with the intra-operative CT scan, the recording trajectory was confirmed and corrected whenever necessary (corrections were always <1 mm). The distance of the microelectrode from the ventral STN target was measured using our microdrive. The location of each recording was identified as the point along the trajectory with the measured distance from target. In the case of multiple simultaneously used microelectrodes, the additional trajectories were offset by the distance of electrodes (2 mm) in the appropriate direction. These calculations were performed using the Brainlab Elements stereotaxy software.

## Behavioral task

After identification of a stable single or multiunit, we proceeded with our behavioral task. A monitor on a jointed arm was positioned in front of the patient. A stereoscopic infrared camera (Leap Motion Controller, San Francisco, CA) was positioned under the subject's contralateral hand with respect to the recorded STN. The subject was instructed to rest his or her elbow on the armrest and make forearm movements to control the cursor. The subject's hand, therefore, was assumed to move on the surface of a sphere with elbow at its center. The built-in software of the Leap Motion Controller identified the absolute position of the subject's palm relative to the controller (sampling rate ~100 Hz). The device latency was ~10 ms. The palm position was mapped to a spherical surface with center located at approximately the subject's elbow and radius equal to the forearm length. The horizontal and vertical angles of this position were scaled to generate the x- and y-coordinates of the cursor on the screen.

A central white circle (fixation point) appeared on the screen at the beginning of each trial. The subject moved the cursor into the fixation point to indicate readiness, beginning the fixation period. After a variable duration (truncated exponential distribution, range, 300–600 ms, mean 400 ms), an instruction cue appeared on the right or left of the screen, which indicated both the type and the direction of the trial. A green circle indicated a Reach trial (60–70% of trials), instructing subjects to move the cursor to a target area defined by the location of the cue. A blue arrow indicated a Planned Turn trial (30–40% of trials), instructing subjects to move toward the cue and then upward to a target area at the top of the screen. A minimum horizontal distance from the fixation point had to be attained before the turn (i.e., subjects could not make a straight, diagonal movement to the target region). Subjects kept their hands still to maintain the cursor in the fixation area until the fixation point disappeared (go cue; truncated exponential delay, range 400–1000 ms, mean 600 ms). They were instructed to initiate their movements as soon as they detected the go cue. Trials on which subjects left the fixation area before the go cue were halted and treated as fixation breaks. On a random half of the trials that were initially cued as Reach trials, the cue changed to a blue arrow during the movement toward the target (turn cue; 43–67% of Reach trials, 30–40% of total trials). The cue changed when subjects entered a rectangular window whose center was chosen from a uniform distribution that spanned 35–65% of distance between the fixation and target points. Subjects had to change their movement paths, making a vertical movement to a target region at the top of the screen. These

were designated as Impromptu Turn trials. After completion of each trial, distinct auditory feedback indicated correct responses, incorrect responses (trajectories that did not meet the criteria above), or fixation breaks. The inter-trial interval was 1.5 s.

A session was terminated when the subject showed signs of fatigue or after approximately 10 min. If subjects wished to perform a second block of the task, the microelectrode was moved to a new location with stable units and a new block started. Surgical implantation of DBS electrodes then proceeded per our usual protocol.

## Spiking data analysis

Spike detection and sorting were performed using an offline sorter (Plexon, Dallas, TX) on raw electrophysiological recordings sampled at 44 kHz. Spike-sorting was performed manually using principal components of spike waveforms. Single units, multi-units, and unsorted background spiking (hash) were considered as separate units. The characteristics of all units are shown in *Supplementary file 2*. Single units were those with the clearest isolation from background and most consistent action potential waveforms (8/39 units). They had a baseline median firing rate of 32.9 Hz (IQR: 18.3–53.9), consistent with single units identified in past studies (*Sharott et al., 2014*; *Weinberger et al., 2006*). Furthermore, the median firing rate of units that were not well-isolated was 40.4 Hz (IQR: 20.7–82.6), consistent with the results of Weinberger et al., suggesting that most multi-units are largely contributed to by a single neuron. The baseline firing rate was calculated from spikes in the intervals between trials. Similar conclusions were reached using well-isolated units only.

All analyses were performed using Matlab R2017a. Spikes were aligned to behavioral events. In addition to events defined by the stimuli on the screen, we also identified two events based on hand movements on each trial. Movement onset was defined as the period of maximum acceleration that most closely preceded the peak velocity of cursor. Turn onset was found as the time of maximum angular acceleration of the cursor. The calculated movement and turn onsets were examined visually relative to each movement trajectory and manually corrected as needed by author D.L. Manual corrections of these events, if needed, were small and blinded to the electrophysiological data and the trial type.

To compare spiking activity around the turn cue on Impromptu Turn trials with spiking activity on Reach and Planned Turn trials, we sampled randomly from the distribution of turn cue onsets on Impromptu Turn trials to define corresponding analysis windows on Reach and Planned Turn trials. On Reach trials, subjects did not make a turn movement, and therefore, there was no turn onset event. To compare responses around the onset of the turn movement on Planned and Impromptu Turn trials with Reach trials, we sampled randomly from turn onset times on Planned Turn and Impromptu Turn trials to define corresponding analysis windows on Reach trials.

We sought to determine spiking activity associated with the variety of task events. Inter-event intervals were variable because they were selected from random distributions or depended on the subject's responses. In calculating PSTHs aligned to an event, we censored spikes that occurred before the preceding event or after the following event. Spikes occurring within 500 ms of the end of the preceding trial were also censored. For smoothing the PSTHs, we used a causal kernel (alpha function, $\alpha^2 \tau e^{-\alpha \tau}$ for $\tau > 0$ and 0 for $\tau \leq 0$, with $\alpha = 20s^{-1}$). PSTH error bars were calculated as the standard error over trials. Units were excluded from analyses if there were fewer than four correct trials of each type or if the subject was unable to perform the task (mean ± s.d., 21.8 ± 8.1 trials per trial type per subject). There was a mean ± s.d. of 18.8 ± 11.9 trials per subject split across three conditions that did not meet the accuracy criteria. Adding these trials to the analyses did not critically change our results. The number of correct trials of each type for each unit is shown in *Supplementary file 2*. For analysis of PSTHs on medial and lateral trials, we only examined units with at least four correct medial and lateral trials of each type (mean ± s.d., 13.5 ± 3.3 trials per trial type per subject). We had planned to collect data from approximately 25 units in this study as this would allow us to detect a difference of 0.5 standard deviations between conditions assuming pairwise comparisons and a Holm-Bonferroni corrected p-value of 0.05 needed for significance. We had expected to record 3–4 units per STN resulting in a sample size of 9 STNs. We recorded from an average of 1.95 units per recording location. 62.5 % of subjects (5/8) had multiple recording sites.

The analysis window aligned to each task event was chosen such that at least 2/3 of trials of each subject contributed to the analysis at each point in time. Because fixation, instruction, and feedback

events were separated from their neighboring events by large intervals, their analysis windows were wide (fixation, [−500 ms, + 300 ms]; instruction, [−200 ms, + 300 ms]; feedback [−230 ms, + 1000 ms]; windows are with respect to the corresponding event times). These large windows were adopted to maximize our precision for estimation of firing rates, but similar results were obtained with shorter windows too. Since movement and turn onsets had short latencies relative to the Go and Turn cues, analysis windows aligned to these events had to be narrower (movement onset, [−300 ms, + 270 ms]; turn onset, [−270 ms, + 220 ms]).

To explore the similarity of responses across recorded units, we created a response profile for each unit by concatenating the unit's PSTHs for each trial type in the analysis windows listed in the previous paragraph. To account for different ranges of firing rates across units, the response profiles of each unit were z-scored based on the mean and standard deviations obtained across all trials. Then, we performed PCA on the response profiles across recorded units to find the key patterns that shaped the profiles. As there was no turn onset event on Reach trials, the principal components were determined based on Planned and Impromptu Turn trials only. Reach trial response profiles were then projected onto this space. The top four principal components explained 61 % of the total variance on Planned and Impromptu Turn trials, indicating that a small number of patterns captured the diversity of neural responses. The coefficients of the top four components were used to perform a hierarchical clustering analysis on the recorded units using the Chebyshev distance metric and a complete linkage function. The analysis revealed two main clusters that accounted for 87 % of units. The remaining units were part of a third cluster with strong responses around planned turns and activity profiles that resembled those of the movement-selective cluster of units. However, we could not extensively explore this third cluster due to low sample size.

## Statistical analysis

Significant differences between firing rates across different trial types were assessed by averaging firing rates within time intervals of interest and making pairwise comparisons using a linear mixed effects model with a fixed effect for the difference of firing rates and random effects for subjects and for units nested within subjects. When reporting pairwise comparisons in cases where there are more than two groups, the p-values were corrected using the Holm–Bonferroni method.

Choosing analysis intervals in time series data is susceptible to Type I error as one can often define a large number of potential analysis intervals within an epoch of interest. It would therefore be ideal to have a systematic approach that guards against such errors while also maintaining adequate temporal resolution to capture transient effects in the data. We developed such an approach (available at https://github.com/dlondon12/InterruptedReachClustMass); copy archived at SWH swh:1:rev:c597f70832fa3bc86e53aaa6030cba643e60f92b ;*London, 2021*),, for our dataset by adapting the between-subjects cluster mass test (*Maris and Oostenveld, 2007*). The cluster mass test allows detection of significant differences over contiguous time points while controlling for the family-wise error rate, addressing the multiple comparisons and low temporal resolution problems in conventional tests.

Traditionally, two groups are compared in the cluster mass test. However, since most of our analyses concerned differences among three groups (i.e., the different trial types), we extended this method. The cluster mass test consists of two basic operations: selecting the time clusters and testing their significance. In the original method, two groups are compared at each point in time to generate a *t*- or *z*-statistic that is thresholded to select time clusters at a desired false discovery rate. Instead of calculating a *t*-statistic for two groups, we calculated an *F*-statistic for three groups at each point in time, and then thresholded it to select our clusters of interest.

In our analyses, we consider units to be random effects and trial types to be fixed effects (analogous to a repeated measures ANOVA). The *F*-statistic was calculated as follows. For each unit $j$, consider a firing rate response profile, $r_{j,m}(t)$, where $m$ represents the trial type. Let us denote the mean response across units with $\bar{r}_m(t)$, across trial types with $\bar{r}_j(t)$, and across units and trial types with $\bar{r}(t)$. The total sum of squares, $SS_{tot}$, the sum of squares across the $M$ trial types, $SS_M$, and across the the $N$ units, $SS_N$, are:

$$SS_{\text{tot}}(t) = \sum_{m=1}^{M}\sum_{j=1}^{N}\left(r_{j,m}(t) - \bar{r}(t)\right)^2$$

$$SS_M(t) = N\sum_{m=1}^{M}\left(\bar{r}_m(t) - \bar{r}(t)\right)^2 \tag{1}$$

$$SS_N(t) = M\sum_{j=1}^{N}\left(\bar{r}_j(t) - \bar{r}(t)\right)^2$$

Considering each unit as a repeated measure, the residual sum of squares is

$$SS_{\text{error}} = SS_{\text{tot}} - SS_M - SS_N \tag{2}$$

and the statistic for the difference across trial types is

$$F_{M-1,(N-1)(M-1)}(t) = \frac{\dfrac{SS_M(t)}{M-1}}{\dfrac{SS_{\text{error}}(t)}{(N-1)(M-1)}} \tag{3}$$

This statistic was thresholded at the 90th percentile of the *F*-distribution with parameters $M-1$ and $(N-1)(M-1)$ . Contiguous blocks of test statistics above threshold were considered to be time clusters. We then calculated a statistic for each cluster, termed the cluster statistic. Among the commonly used statistics, we chose the sum of $SS_{\text{error}}(t)$ within each cluster in this paper, but we also obtained similar results with other relevant statistics including the sum of $SS_M(t)$ or of $F_{M-1,(N-1)(M-1)}$ . The null hypothesis is that the $M$ trial types are no different and thus interchangeable. We therefore created our null distribution by permuting the labels on each trial type *within* each unit, calculating the *F*-statistic for each permutation to select clusters, and determining the maximum cluster level test statistic. The cluster level statistics for the real data were compared with this null distribution to calculate p-values.

Because our framework is analogous to a repeated measures ANOVA with a single factor (e.g. trial type) per unit, it can also be generalized seamlessly to a repeated measures ANOVA with two factors (e.g. trial type and movement direction). For this extension, consider the additional factor $s$, with $S$ levels. Our goal is to determine significant time clusters of each main effect, while controlling for interaction effects. For each combination of the two factors, each unit has a different firing rate profile, $r_{j,m,s}(t)$ . For factor $m$, the sum of squares is

$$SS_M(t) = NS\sum_{m=1}^{M}\left(\bar{r}_m(t) - \bar{r}(t)\right)^2 \tag{4}$$

For factors $m$ and $s$, we must also consider the interaction sum of squares:

$$SS_{M,S}(t) = N\sum_{m=1}^{M}\sum_{s=1}^{S}\left(\bar{r}_{m,s}(t) - \bar{r}_m(t) - \bar{r}_s(t) + \bar{r}(t)\right)^2 \tag{5}$$

where $\bar{r}_{m,s}(t)$ is the mean across trials for the factors $m$ and $s$. Analogous expressions can be constructed for each individual factor and each pair of factors. *F*-statistics are calculated for each main effect:

$$F_{M-1,(N-1)(M-1)}(t) = \frac{\dfrac{SS_M(t)}{M-1}}{\dfrac{SS_{M,N}(t)}{(N-1)(M-1)}}$$

$$\tag{6}$$

$$F_{S-1,(N-1)(S-1)}(t) = \frac{\dfrac{SS_S(t)}{S-1}}{\dfrac{SS_{S,N}(t)}{(N-1)(S-1)}}$$

These *F*-statistics are thresholded to generate clusters. The cluster-level statistic is the sum of the sum of squares in the denominator of the respective *F*-statistic calculation. For each tested effect, there is a separate permutation distribution depending on the 'exchangeable units' (*Anderson, 2001*). For testing factor $m$, the levels of $m$ are permuted within each unit and without permutation

of the other factor. The remainder of the calculation continues as for the one factor case, resulting in separate p-values for each main effect, as in a multi-way ANOVA.

We further extended this framework to the case of multidimensional data, as in the case of principal components analysis. Up until this point we have considered the case of $N$ units which represent repeated measures. Principal components analysis of these units results in dimensionality reduction to $D$ dimensions. These dimensions are *not* repeated measures; they represent orthogonal components of the data. Therefore, instead of considering one-dimensional repeated measures data of the form $r_{j,m,s}(t)$, we consider $D$-dimensional data of the form $Q_{m,s}(t) = [q_{1,m,s}(t), \cdots, q_{D,m,s}(t)]$, where $Q_{m,s}(t)$ is the projection of $r_{j,m,s}(t)$ onto the first $D$ principal components, and $q_{d,m,s}(t)$ is the projection onto the $d$-th principal component.

Our goal is to find the times at which the trajectories in the PC space are significantly different. As there are no repeated measures, we cannot calculate *F*-statistics, but we can use the sum of squares directly. Again, we analyze each factor separately: for each trial type $m$, we consider $\overline{Q}_m(t)$, where each time point is the centroid across the levels of $s$, and $\overline{Q}(t)$, where each time point is the centroid across the levels of $m$ and $s$. We calculate the sum of squares by adding the squared Euclidean distances from the centroid by analogy with *Equation 4*:

$$SS_M = S \sum_{m=1}^{M} \left\| \overline{Q}_m(t) - \overline{Q}(t) \right\|^2 \tag{7}$$

where $\|.\|$ is the Euclidean norm.

We threshold these sums of squares above the 90th percentile as was done with the *F*-statistics. To determine the value of the 90th percentile, we generate a permutation distribution as follows. For each unit, we randomly permute the trials across the appropriate factor without permutation of the other factors. The mean firing rates are calculated for each unit across the permuted trials, and the population activity patterns projected on the same principal components as the original dataset. These projections are then used to calculate the sums of squares. After creation of the clusters, the cluster-level statistics are calculated by adding the sum of squares values within each cluster. The maximum cluster-level statistic on each permutation is used to generate the distribution of cluster level statistics across permutations. The p-values for clusters of the original dataset are calculated as the probability of values in the permutation distribution being higher than the cluster level statistics.

Once significant clusters are located with this multidimensional test, we perform pairwise comparisons of the relevant conditions to determine if the distances in state space between conditions in these time intervals are larger than those expected by chance. At each time point, we calculate the mean squared Euclidean distance between the trajectories of levels 1 and 2 of factor $m$, $\frac{1}{\Delta t} \sum_t \left\| \overline{Q}_1(t) - \overline{Q}_2(t) \right\|^2$. To generate the null hypothesis distribution for comparison, we calculate this same statistic based on the permutation procedure explained above. If factor $m$ has only two levels, the projections of permutated data on principal components are the same as those generated above for the cluster mass calculation. If factor $m$ has more than two levels, we generate new permutation projections on the principal components by permuting trials as above but only for the levels being compared. The mean squared Euclidean distance is compared to this distribution to calculate p-values. The mean squared Euclidean distance is subtracted from each value in this distribution to generate a new distribution, which is then used to calculate the mean and 95 % confidence interval of the mean squared Euclidean distance above chance-level distance (*Figure 6J–K*). A sufficient number of permutations was used for these tests to allow for a minimum Holm-Bonferroni corrected p-value of 0.001 (1000 iterations for a single pairwise comparison and 3000 iterations for three comparisons).

Finally, we used this same framework to analyze firing rate differences between trial types within individual units. Here, we focus on the firing rate on each individual trial instead of the PSTH across all trials for the unit. Each trial contains only a single condition (e.g., Planned Turn). The single-trial firing rate at time $t$ on trial $i$ of condition $m$ is $r_{i,m}(t)$, resulting in the following sums of squares:

$$
\begin{aligned}
SS_{\text{tot}}(t) &= \sum_{m=1}^{M} \sum_{i=1}^{N_m} \left( r_{i,m}(t) - \overline{r}(t) \right)^2 \\
SS_M(t) &= \sum_{m=1}^{N_m} \left( \overline{r}_m(t) - \overline{r}(t) \right)^2 \\
SS_{\text{error}} &= SS_{\text{tot}} - SS_M
\end{aligned}
\tag{8}
$$

where $N_m$ is the number of trials on condition $m$. The F-statisic is

$$F_{M-1,N-M}(t) = \frac{\frac{SS_M(t)}{M-1}}{\frac{SS_{\text{error}}(t)}{N-M}} \tag{9}$$

where $N$ is the total number of trials. The 90th percentile of the *F*-distribution is used to select clusters whose cluster-level statistic is the sum of $SS_{\text{error}}$ as in the repeated measures case. These cluster statistics are compared to those generated using a null distribution created by randomly partitioning the trials among the conditions keeping the trial count in each condition constant. This comparison results in a p-value for each cluster.

We used this method to quantify the number of units that reflect the firing rate differences identified on the population level. If a time interval contained a significant difference between a pair of conditions across the population of units, we determined if this time interval contained a significant difference between that pair of trial types within each unit. A significant difference was considered to be a cluster with p<0.1. Under this criterion, we would expect 10 % of units to contain significant differences by chance alone; this is the null hypothesis. We calculated a p-value for this null hypothesis as the probability of obtaining at least as many significantly different units under the binomial distribution with success rate of 10 % (this is the binomial test).

## Mixed effect models

We tested if the firing rate of the recorded units changes as a function of the movement speed and acceleration. During trial , the hand position was used to update the cursor position $(x_i(t), y_i(t))$ on the screen. The cursor position was filtered using a Gaussian kernel (10 ms standard deviation) and used to calculate instantaneous speed and acceleration at time $t$:

$$\begin{aligned} \left(v_{x,i}(t), v_{y,i}(t)\right) &= \left(\frac{x_i(t) - x_i(t-dt)}{dt}, \frac{y_i(t) - y_i(t-dt)}{dt}\right) \\ \left(a_{x,i}(t), a_{y,i}(t)\right) &= \left(\frac{v_{x,i}(t) - v_{x,i}(t-dt)}{dt}, \frac{v_{y,i}(t) - v_{y,i}(t-dt)}{dt}\right) \end{aligned} \tag{10}$$

where $dt$ is the time interval between consecutive measurements. We then calculated the magnitude of the instantaneous speed and acceleration vectors on the screen:

$$\begin{aligned} v_i(t) &= \left\| \left(v_{x,i}(t), v_{y,i}(t)\right) \right\| \\ a_i(t) &= \left\| \left(a_{x,i}(t), a_{y,i}(t)\right) \right\| \end{aligned} \tag{11}$$

where $\|.\|$ is the Euclidean norm. To combine data across sessions, we normalized $v_i(t)$ and $a_i(t)$ by z-scoring across all trials in each session. *Figure 4* shows averages of normalized speed and acceleration for each trial type, $m$, aligned to different task events.

To evaluate the relationship between the firing rate of each unit and movement kinematics, we used a lagged regression analysis:

$$\bar{r}_m(t) = \beta_0 + \beta_1 \bar{v}_m(t + \Delta t) + \beta_2 \bar{a}_m(t + \Delta t) \tag{12}$$

where $\bar{r}_m(t)$ is the mean firing rate of the unit at time $t$ on trial type $m$, $\bar{v}_m$ and $\bar{a}_m$ are the mean movement speed and acceleration, respectively, and $\Delta t$ is the time lag. The $\beta$ parameters were fit as separate fixed effects for each unit. We systematically varied $\Delta t$, searching for the model with the highest R² as the best fitting model. Across units and across $\Delta t$ that were modeled, the firing rate time points that were fitted were kept constant. The maximum positive and negative $\Delta t$ determined which data points were fitted and which were held out for model testing. Consider an $\bar{r}_m(t)$ on the range $\left[t_{\text{start}}, t_{\text{end}}\right]$ and modeled on the range $[\Delta t_{\min}, \Delta t_{\max}]$. There is a subset of points in $\left[t_{\text{start}}, t_{\text{end}}\right]$, such that there is data for $\bar{v}_m(t + \Delta t)$ and $\bar{a}_m(t + \Delta t)$ (i.e. $t + \Delta t$ is also on the interval $\left[t_{\text{start}}, t_{\text{end}}\right]$). Therefore, $\bar{r}_m(t)$ was only fit on the range $\left[t_{\text{start}} - \Delta t_{\min}, t_{\text{end}} - \Delta t_{\max}\right]$. The optimal value of $\Delta t$ determined which time points were on this range and were, therefore, used to fit the model. The time points that were not fit but were on the range $\left[t_{\text{start}} - \Delta t, t_{\text{end}} - \Delta t\right]$ (note this range contains the best fit ) were used as a testing set for the models. *Equation 12* was chosen as a good descriptive model of firing rates across all units after

extensive search of possible linear models of speed and acceleration. Confidence intervals for $R^2$ were calculated using bootstrapping (1,000 iterations).

Overall, we compared 14 alternative models that contained a combination of speed, acceleration, and their interaction (*Figure 4—figure supplements 3–4*). In some models, the x- and y-components of speed and acceleration in the direction of target or their absolute values were included instead of the total magnitude of speed and accleration. Our critical results were replicated in the alternative models that had the top out-of-sample fits to the data.

## Ideal observer analysis

To determine whether population firing rate differences were sufficient for decoding of behavior, we used a cross-validated ideal observer analysis to determine if the firing rate changes across the population of units were sufficient to correctly classify the trial type. Our procedure is as follows. On each iteration, we randomly held out one trial of the same type from each unit (test set). Next, we calculated the mean response of each unit on the rest of the trials (training set) marginalized by the trial type. We then calculated the mean Euclidean distance of the firing rate patterns of the held out trials across the population from each of the patterns of marginalized mean responses for the training trials. The held out trials were classified according to the smallest Euclidean distance. We performed 10,000 iterations, randomly dividing the trials for each unit into training and test sets. The decoding accuracy was the fraction of iterations correctly classified.

## LFP and joint LFP-spiking analysis

For LFP analyses, the raw 44 kHz recorded voltage signals from each electrode were downsampled to 2 kHz. We then locally detrended and removed line noise using the Chronux toolbox (*Mitra and Bokil, 2008*), augmented with manual artifact removal where needed. The LFP from 2.5 s before to 2.5 s after an artifact was censored from further analysis. We then calculated the continuous wavelet transform using a Morse wavelet with symmetry parameter of 3 and time-bandwidth product of 20. We aligned the wavelet-transformed data across trials for each session, averaged across all trials of the same type (in the complex plane), and calculated the squared magnitude to determine the evoked power. To calculate the induced power, we computed the squared magnitude of the aligned wavelet-transformed data, averaged across trials, and subtracted the evoked power.

Evoked and induced power calculations are sensitive to varying trial counts. In our study, each condition had 4–47 trials depending on the session. Furthermore, because of variability in the timing of events in our task, the number of trials available for analysis varied by time. To control for potential noise induced by this variability, we used the time period between 500 ms to 250 ms before the fixation onset to calculate a baseline for the average evoked and induced power for each trial. We then used a permutation procedure where we sampled without replacement from trials in each session, without regard to trial type, in order to estimate the average powers for a variety of trial counts. Repeating this process one hundred times generated a baseline distribution of evoked and induced power for any trial count at each time point. These baselines were separately calculated for each recording session. We normalized our calculated evoked and induced power values at each point in time by dividing by the mean of the appropriate baseline distribution for the trial count of that particular trial type at that particular point in time in each session.

We also calculated evoked and induced power of the beta frequency range. To do this we applied a notch filter between 13 and 30 Hz to the downsampled, detrended voltage data. We then used the Hilbert transform to calculate the analytic signal. Evoked and induced power were calculated from this signal and normalized by the baseline for the trial counts in the session.

The coupling of spiking activity to the LFP was quantified using the pairwise-phase consistency metric (PPC). We use the $\hat{P}_2$ metric described by *Vinck et al., 2012* which is not vulnerable to biases caused by varying firing rates or dependencies between the firing rate and the distribution of spike phases. Briefly, we used the continuous wavelet transfrom to calculate the phase of each spike at every frequency of the LFP, termed the spike-triggered phase. The spikes were then aligned to task events and segmented into bins. For each time point, PPC was calculated using spikes that preceded that point in time by no more than 10 cycles; bin size, therefore, varied with different frequencies. We capped the bin size at 1.5 s, and spaced time points by 50 ms. We randomly selected a pair of trials of the same type (e.g. two Reach trials). For each pair of spikes in the same bin across the two trials, we

calculated the dot product of spike-triggered phases and averaged the results across the two trials. These averaged dot products were averaged over all pairs of trials of the same type to calculate PPC for a particular bin. This was repeated for all bins allowing us to calculate a spectrogram for PPC. The PPC for the entire beta range was calculated in the same way using the the spike-triggered phase from the analytic signal of the beta-filtered LFP instead of the continous wavelet transform.

We examined the data for differences between spectrograms using the cluster mass test in the same fashion as explained above for firing rates. No significant clusters were located. As explained in the section on Statistical Analysis, we avoided analyzing arbitrarily selected intervals to control for Type I error.

Because the shortest of our event-aligned analysis intervals was approximately 500 ms, we limited our frequency-based analysis to above 4 Hz.

## Movement trajectory analysis

To confirm that subjects performed the task as instructed, we generated average movement trajectories across trial types. Since movements were self-paced and could take different times, averaging the hand position across trials at a particular time would not accurately reflect the variability of trajectories across space. To calculate the mean and standard deviation of movement trajectories in a particular trial type, we devised the following analysis using pairs of trajectories. The mean of an arbitrary vector $\{x_k\}$ is:

$$\mu = \frac{1}{N} \sum_{k=1}^{N} x_k$$

which can be rewritten as the mean of the mean of independently-sampled pairs of data points, $i$ and $j$:

$$
\begin{aligned}
\mu &= \frac{1}{2N} \sum_{k=1}^{N} x_k + \frac{1}{2N} \sum_{k=1}^{N} x_k = \frac{1}{2N} \left( \sum_{i=1}^{N} x_i + \sum_{j=1}^{N} x_j \right) \\
&= \frac{1}{N^2} \sum_{i=1}^{N} \sum_{j=1}^{N} \frac{x_i + x_j}{2}
\end{aligned}
\tag{13}
$$

To calculate the mean of an arbitrary number of trajectories, we used dynamic time warping (*Sakoe and Chiba, 1978*) to align each pair of trajectories, and took the mean of the aligned positions. We then selected 200 data points from this mean over equally spaced time intervals. This is the pairwise mean trajectory. Finally, we calculated the mean across pairwise mean trajectories to calculate the mean trajectory.

The variance can be calculated from pairs of trajectories in a similar way:

$$
\begin{aligned}
\sigma^2 &= \frac{1}{N} \sum_{k=1}^{N} \left( x_k - \frac{1}{N} \sum_{l=1}^{N} x_l \right)^2 = \frac{1}{N} \sum_{k=1}^{N} \left( \frac{1}{N} \sum_{l=1}^{N} x_k - \frac{1}{N} \sum_{l=1}^{N} x_l \right)^2 \\
&= \frac{1}{N^3} \sum_{k=1}^{N} \left( \sum_{l=1}^{N} x_k - x_l \right)^2
\end{aligned}
\tag{14}
$$

The method above was used only to calculate the mean and standard deviation of movement trajectories shown in *Figure 1E*.

To classify single-trial movement trajectories, we performed hierarchical clustering of the trajectories using the discrete Frechet distance (*Eiter et al., 1994*) as a dissimilarity metric between pairs of trajectories. We then used Ward's method (*Ward, 1963*) to generate the cluster tree.

## Acknowledgements

The authors would like to thank Gouki Okazawa, and Saleh Esteki for helpful discussions. This work was supported by the Simons Collaboration on the Global Brain (grant 542997), McKnight Scholar Award, Pew Scholarship in the Biomedical Sciences, and National Institute of Mental Health (R01 MH109180).

## Additional information

### Funding

| Funder | Grant reference number | Author |
|---|---|---|
| Simons Collaboration on the Global Brain | 542997 543009 | Roozbeh Kiani |
| McKnight Scholar Award | | Roozbeh Kiani |
| Pew Scholarship in the Biomedical Sciences | | Roozbeh Kiani |
| National Institutes of Mental Health R01 | MH109180-01 | Roozbeh Kiani |

The funders had no role in study design, data collection and interpretation, or the decision to submit the work for publication.

### Author contributions

Dennis London, Conceptualization, Data curation, Formal analysis, Investigation, Methodology, Project administration, Software, Visualization, Writing - original draft, Writing - review and editing; Arash Fazl, Conceptualization, Data curation, Investigation, Methodology, Project administration; Kalman Katlowitz, Marisol Soula, Investigation; Michael H Pourfar, Investigation, Resources; Alon Y Mogilner, Investigation, Methodology, Project administration, Resources, Supervision; Roozbeh Kiani, Conceptualization, Methodology, Project administration, Resources, Software, Supervision, Writing - original draft, Writing - review and editing

### Author ORCIDs

Dennis London ⓘ http://orcid.org/0000-0001-8134-2683
Kalman Katlowitz ⓘ http://orcid.org/0000-0001-5568-6343
Alon Y Mogilner ⓘ http://orcid.org/0000-0003-1493-0463
Roozbeh Kiani ⓘ http://orcid.org/0000-0003-0614-6791

### Ethics

Subjects signed informed consent including consent to publish. The study protocol was approved by the NYU School of Medicine Office of Science and Research Institutional Review Board. Study ID: S16-01855.

### Decision letter and Author response

Decision letter https://doi.org/10.7554/eLife.64893.sa1
Author response https://doi.org/10.7554/eLife.64893.sa2

## Additional files

### Supplementary files

• Supplementary file 1. Subject demographics. Demographic characteristics of all patients and the number of single units and multi-units recorded from each. Our conclusions do not change if subjects 1, 2, or seven who contributed relatively more units than others are removed from the dataset.

• Supplementary file 2. Unit characteristics. Characteristics of all recorded units including the number of each trial type recorded, the classification by unit type, and the baseline firing rate (inter-trial interval). The median baseline firing rate of single- and multi-units was 32.9 Hz (IQR: 18.3–53.9) and 40.4 Hz (IQR: 20.7–82.6), respectively, suggesting that most spikes recorded from multi-units originate from one single unit. Twenty of these units had a sufficient number of left-sided and right-sided trials for independent PSTHs to be calculated for these conditions. Analyses of firing rates marginalized on trial direction (*Figures 5 and 6*, and *Figure 5—figure supplement 2*) use these units.

• Transparent reporting form

## Data availability

All raw and processed data has been uploaded to Dryad. DOI:https://doi.org/10.5061/dryad.2jm63xsq2.

The following dataset was generated:

| Author(s) | Year | Dataset title | Dataset URL | Database and Identifier |
|---|---|---|---|---|
| London D, Fazl A, Katlowitz K, Soula M, Pourfar M, Mogilner A, Kiani R | 2021 | Distinct population code for movement kinematics and changes of ongoing movements in human subthalamic nucleus | https://doi.org/10.5061/dryad.2jm63xsq2 | Dryad Digital Repository, 10.5061/dryad.2jm63xsq2 |

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
