## [Decision Letter]

**Acceptance summary:**

This manuscript reports on a unique data set of human subthalamic nucleus electrophysiological recordings during a behavioral task. The results (which include single unit, multi-unit, and local field potential activity) point to the complex nature of the neural code in this structure and challenge some existing models related to the role of the subthalamic nucleus.

**Decision letter after peer review:**

[Editors’ note: the authors submitted for reconsideration following the decision after peer review. What follows is the decision letter after the first round of review.]

Thank you for submitting your work entitled "Distinct population code for movement kinematics and changes of ongoing movements in human subthalamic nucleus" for consideration by *eLife*. Your article has been reviewed by 3 peer reviewers, and the evaluation has been overseen by a Reviewing Editor and a Senior Editor. The following individuals involved in review of your submission have agreed to reveal their identity: Andrew Sharott (Reviewer #1); Harrison Walker (Reviewer #3).

Our decision has been reached after consultation between the reviewers. Based on these discussions and the individual reviews below, we regret to inform you that your work will not be considered further for publication in *eLife*.

While the reviewers were overall positive about many aspects of the manuscript, there were a number of points they brought up. Upon discussion, we felt that the changes required were substantial enough that we were not confident they could be fully addressed in a revision. In particular, the reviewers noted missing information about the type of patient(s) studied and noted that this could have a substantial impact on the interpretations of the results. Additionally, we felt that important information about number of units contributed by each patient (and how those may have influenced different STN sub-populations was not clear). Finally, the reviewers felt additional statistical analyses and justification were necessary. While the reviewers agreed that it might be possible for all these points to be addressed in a revision, they also noted it may substantially change the interpretation (depending on the updated results). The full reviewers are included below.

*Reviewer #1:*

1. There are important details missing regarding the data and data acquisition. What type of patients were these recordings made in? (a) Are they from different patient populations? This is an important consideration for several aspects of the study including the clustering of neurons into different response types and the analysis of β oscillations. (b) What is the baseline firing rates of the STN neurons and how many were considered single units? The homogeneity of STN unit spiking means that single units should be in consistently firing at a relatively high rate (10-35 spikes/s). Defining the entrance, the STN alone is not sufficient to establish that the units in the STN (particularly at the ventral border). Further evidence that the units were in the STN and were single units is important to substantiate the authors claims as to the relevance of their results to STN function.

2. On a related issue, do the authors have information as to the spatial parameters of the STN recordings? If so, do these parameters correlate with the function clustering of the response types?

3. The authors use custom use a range of custom designed data analyses to tease out STN neurons with tendencies to respond to differently to parts of the task in the different conditions. In some cases, this is a strength of the study and many of the analyses elegantly pull out features of the data that may not be immediately clear (e.g Figures3 and 4). However, in it's current form it is difficult to establish the effect sizes for other analyses – particularly in the case of the "Turn Units." While the authors go to impressive lengths to show that the differences at the Turn are different, from Figure 3C they seem to be very small and a fraction of those seen in response to kinematic parameters. Currently, many of the novel claims are dependent on this difference (Abstract: "We show that STN activity decreases during action switches, contrary to prevalent theories".). Can the authors provide evidence of the effect size of the responses to task switching as well as being able to say they are significantly different? How many individual units are able to distinguish between specific aspects of the task (such as the example in Figure 2C).

4. Given the size of the sample – the authors should provide some evidence that their main findings are scene across the majority of patients. For example, how may units in each cluster are derived from different patients?

5. The analysis of β oscillations/synchrony is potentially very interesting, but I cannot interpret it's relevance without knowing the disease-status of the patients.

*Reviewer #2:*

The authors report on subthalamic nucleus neural population dynamics and its role in movement planning and execution, especially as it relates to changes in the motor plans (they compared trials of straight reach, planned turn or unexpected turn while recording 39 single units from 8 patients undergoing DBS surgery). They used principal component analysis of firing rates and unsupervised clustering to identify two types of cells – movement and turning units, each contributing specific information regarding movement dynamics encoded in the STN (the third identified cluster was too small to analyze). The manuscript is generally well written and the complex analysis is logically presented although sometimes conclusions are difficult to appreciate from the figures. This type of analysis is increasingly necessary to try and understand the complex dynamics in the nervous system, but the challenge will be to relate to behavior in a meaningful way. The authors should relate their findings to (presumably) parkinsonian pathology especially when discussing synchronization of units with β band oscillations which is the hallmark of the disease.

– The analysis is presented only for move and turn onsets. Was there a difference in firing rates at Go and Impromptu Turn cues? Also, when was the impromptu turn cue given and did this differ by movement speed (within and across subjects)?

– "This result was not due to lack of statistical power of the cluster mass test, which showed great sensitivity to systematic trends in our datasets." – It is difficult to assess such statements which occur several times. Could you explain what trends?

– "Neuronal spiking in STN is typically phase-locked to β oscillations (Kühn et al., 2005; Weinberger et al., 2006)" – is this because of parkinsonian state? The authors need to acknowledge that subjects had Parkinson's disease (presumably, this is not stated anywhere) so findings may be influenced by pathological parkinsonian state, especially in regards to single cell firing being phase locked to β oscillations which is known to be exaggerated in PD.

– "Interestingly, these sub-populations are partially spatially segregated within STN, with Movement Units located more laterally…" – provide a depiction of unit locations if commenting on this in the Discussion.

– "Further, stimulation of glutamatergic cortical inputs to STN (i.e., the hyperdirect pathway) worsens bradykinesia (Gradinaru et al., 2009)" – that paper showed quite the opposite – stimulation of hyperdirect pathway improved bradykinesia. Please address in your conclusions.

– "To compare spiking activity around the turn on the Impromptu Turn trials with spiking activity on Reach and Planned Turn trials, we sampled randomly from the distribution of turn cue onsets on Impromptu Turn Trials to define corresponding analysis windows on Reach and Planned Turn trials."

– I understand why you had to do this for Reach trials, but why did you not use the actual turn on Planned turn trials as the analysis window?

– Methods: "Subjects were patients undergoing implantation of deep brain stimulation…" Please specific that these were presumably patients with Parkinson's disease, and were they off dopaminergic medications?

– Authors report "13.5{plus minus}3.3 trials per trial type per subject" – How may trials of each type were done for each unit recorded? How many units were recorded per patient?

*Reviewer #3:*

Conceptually, even if 'impromptu' units are an actual subpopulation of neurons, a decrease in STN activity just before the execution of the unexpected turn seems consistent, not inconsistent, with previous hypotheses on STN as a "brake" or "global stop" nucleus. I think they could clarify their thoughts on this more in their writing.

Would like to see more specific examples of sorted units and examples that clearly demonstrate the phenomena they are claiming.

All group level plots and panels should include the number of unique participants, the number of individual units, and the number of trials. For example, in Figure 2e PC1 appears to just be the PTH from the unit in Figure 2C. This unit appears to overwhelmingly dominate the population variance. Was PCA performed across all subjects/trials/units?

Given 39 units / 9 STNs = 4 units per STN. How many subjects had multiple recording sites, what was the average number of units per recording site?

Plots should generally be aligned to impromptu turn cue onset. Without this time alignment, it’s unclear whether the observed changes are related to the cue, the prior movement, or the subsequent movement. Even with this alignment there are some questions about prior events spilling into the 'impromptu' turn condition, depending on the latency of the second command and the nature of the kinematic response of the units in a given experiment.

Figure 2B/D: How do these trajectories differ? Following feedback, the traces look identical. Were stats performed across all principal components?

Figure 3C: Turn versus impromptu turn changes are < 0.2 Z-scores – these are not convincing changes.

Use of terms ipsilateral and contralateral is confusing. I think they mean medial/lateral or flexor/extensor as these tasks appear to be occurring in the same hand.

Figure 7E-F – description says that turn trials show a decreased spike synchronization, but this is because β disappears. Also it appears that this happens regardless of trial type (not just during turns). They might consider also including high frequency LFP activity for comparison with β.

The authors should explicitly state what disease the diagnoses of the patients (assuming Parkinson's disease in all, since these are STN implants).

Supplemental Video with 3D PCA visualization, why are the plotted dimensions PC1, PC3, and PC4? Where is PC2?

[Editors’ note: further revisions were suggested prior to acceptance, as described below.]

Thank you for resubmitting your work entitled "Distinct population code for movement kinematics and changes of ongoing movements in human subthalamic nucleus" for further consideration by *eLife*. Your revised article has been evaluated by Richard Ivry (Senior Editor) and a Reviewing Editor.

The manuscript has been improved but there are some remaining issues that need to be addressed, as outlined below:

Primarily we had some concerns about the interpretation of the model especially with response to the "Turn units" and what they may actually be encoding. We have further elaborated on this concern, as well as highlighting a few additional concerns below.

Essential revisions:

1) Please comment more on what the 'turning' units are actually encoding, and if they could actually be encoding the acceleration and velocity in the vertical axis? Further, do the results still support the statement that 'one sub-population encoding movement kinematics and direction and another encoding unexpected action switches'. Specifically, Figure 4 indicates that both 'Movement Unit' and 'Turn Unit' firing rate can reflect kinematic parameters with different time lags. The authors argued that (Line 276-278): '… the strength of this relationship (for Turn Units) was considerably weaker than for Movement Units (mean R2 = 0.45 compared to 0.61 for Movement Units). Furthermore, movement kinematics did not accurately predict the PSTHs of these units (Figure 4 —figure supplement 2B, D, and F).' However, it is quite subjective to say the R2 values are 'considerably weaker'. Comparing the plots of A, C and E against B, D, and F in Figure 4 —figure supplement 2, can the author's confidently say that movement kinematics can predict Movement Units (shown in plots of A, C and E), but cannot predict Turn Units (shown in plots of B, D and F)? Furthermore, even if the model fitting is a bit worse for Turn Units compare to Movement Units, did the authors used the same model that fits best for movement units? Or to be more fair, did the authors do model comparison for best fitting models for the two types of Units separately?

2) The authors mentioned that 'Turn Units decreased their firing rates before turn onset on Impromptu Turn trials but increased them before turn onset on Planned Turn trials.' (Line 289-291). However, we don't see this statement being supported by data/figure. Can the authors provide evidence/data for this? Or did we miss anything?

3) Reviewer 3 from the previous round of review raised that 'Conceptually, even if 'impromptu' units are an actual subpopulation of neurons, a decrease in STN

activity just before the execution of the unexpected turn seems consistent, not inconsistent, with the previous hypotheses on STN as a "brake" or "global stop" nucleus. We think the authors should clarify their thoughts on this more in their writing.' We feel this was not sufficiently addressed.

4) For the statistical analysis Equation. 1 – Equation. 9, it is not clear whether the authors have used self-programmed scripts for these tests or existing toolbox/functions in Matlab to do this. As far as I understand, Equation. 1 – Equation. 9 are similar to what is implemented in the linear mixed effects models available in Matlab (https://www.mathworks.com/help/stats/linear-mixed-effects-models). This can be applied to each time point, and then using cluster mass test to identify the time window with significant effect. Or alternatively, can the authors specify what is difference between the method used in this study and the linear mixed effects model? The authors mentioned that 'For testing factor ;, the levels of ; are permuted within each unit and without permutation of the other factor. The remainder of the calculation continues as for the one factor case, resulting in separate p-values for each main effect, as in a multi-way ANOVA.' Is it equivalent to repeating the multilevel modeling permuting the levels of m? If the method is based on the framework of multi-level modeling, it might not be necessary to present all the Equations.

5) The accurate rate of different conditions was around 0.7. Did the authors only focus on the 'accurate trials' which satisfy the 'accuracy criteria' or considered all trials? Is there any difference between the neural activities for 'accurate trials' vs 'inaccurate trials'?

7) Can the authors compare the baseline spiking rate of the 'movement related' and 'turning related' units identified by clustering analysis applied on the PSTHs.

---

## [Author Response]

[Editors’ note: The authors appealed the original decision. What follows is the authors’ response to the first round of review.]

Reviewer #1:1. There are important details missing regarding the data and data acquisition. What type of patients were these recordings made in? (a) Are they from different patient populations? This is an important consideration for several aspects of the study including the clustering of neurons into different response types and the analysis of β oscillations. (b) What is the baseline firing rates of the STN neurons and how many were considered single units? The homogeneity of STN unit spiking means that single units should be in consistently firing at a relatively high rate (10-35 spikes/s). Defining the entrance, the STN alone is not sufficient to establish that the units in the STN (particularly at the ventral border). Further evidence that the units were in the STN and were single units is important to substantiate the authors claims as to the relevance of their results to STN function.

Many thanks for your thoughtful comments, which helped us close the gaps in the paper and improve it in the process. All recordings are from a homogeneous population – patients with medically refractory Parkinson’s disease undergoing surgical implantation of DBS electrodes in the STN. They did not take their anti-Parkinsonian medications on the day of surgery. We regret omitting the description of the disease status. The disease status has now been explicitly noted in the abstract, introduction, results, methods, and discussion. Patient age, pre-operative UPDRS motor score, side of recording, and number of single and multi-units recorded per patient are listed in a new table (Supplementary File 1) which we reference in the results. Our conclusions are not changed if any of subjects 1, 2, or 7, who contributed relatively more units than others, are removed from the data (page 43).

The characteristics of all units are now listed in a new table (Supplementary File 2). We were conservative in classifying single-units and chose only units with clearest isolation from background and consistency of action potential waveforms (8/39). The single units had a baseline median firing rate of 32.9 Hz (IQR: 18.3-53.9), consistent with single units identified in past studies (Sharott et al., 2014, Weinberger et al., 2006). Furthermore, the median firing rate of multi-units was 40.4 Hz (IQR: 20.7-82.6), consistent with the results of Weinberger et al., suggesting that most multi-units are largely contributed to by a single neuron. The baseline firing rate was calculated from spikes in the intervals between trials. We discuss this on page 20.

We analyzed both single- and multi-units as is commonly done (Sharott et al., 2014, Zavala et al., 2017, Lipski et al., 2018), and our analyses do not depend on the units being well-isolated. We have largely consistent tuning across single and multi-units, and because all analyses were performed on Z-scored firing rates, they did not depend on differences in the range of firing rates. Critically, single units individually display characteristics seen in the population of all units. We now show in the revised Figure 3 which neural responses are from single units (white asterisks in Figure 3A). Units in this figure are sorted by proximity in the hierarchical clustering algorithm. If our results do not depend on whether a unit is well-isolated we would expect single units to not be clustered together by the algorithm, as is the case in the figure (discussed on page 6).

Microelectrodes were targeted to the ventral border of the STN based on structural MRI and the AC-PC coordinates (11 mm lateral to midline, 5 mm inferior to the AC-PC plane, and 4 mm posterior to the AC-PC midpoint). However, the recorded units spanned the space between the dorsal and ventral borders of STN. Entrance to the STN was noted based on two criteria: (1) real-time alignment of a CT scan performed at the time of surgery with a high-resolution pre-surgical structural MRI (for a detailed explanation, please see our response to the next comment), and (2) an increase in background spiking and identification of rhythmically firing cells at high firing rates. A decrease in background spiking marked the ventral boundary of STN. In many of our surgeries deeper cells along this trajectory fired rapidly with a relatively constant firing rate, characteristic of SNr cells. This procedure has been previously described in detail by our group (Sterio et al., 2002). We have added these points to the Methods (page 19).

2. On a related issue, do the authors have information as to the spatial parameters of the STN recordings? If so, do these parameters correlate with the function clustering of the response types?

We used the pre-operative high resolution structural MRI with gadolinium, pre-operative high resolution CT scan with the Leksell frame in place, planned trajectory, and intra-operative or post-operative CT scan with DBS lead in place to localize our recordings. Our procedure is as follows. The pre-operatively planned target in AC-PC coordinates was identified on MRI. This MRI was merged with the pre-operative frame-localizing CT scan. Using the frame localization, the target was converted into frame coordinates. The pre-operatively planned trajectory was identified using this target and the ring and arc angle used. Based on the alignment of the post-operative CT scan with the DBS lead with the intra-operative CT scan, the trajectory was confirmed and corrected whenever necessary (corrections were always <1mm). The distance of the microelectrode from the ventral STN target was measured using our microdrive. The location of each recording was identified as the point along the trajectory with the measured distance from target. In the case of multiple simultaneously used microelectrodes, the additional trajectories were offset by the distance of electrodes (2 mm) in the appropriate direction. These calculations were performed using the Brainlab Elements stereotaxy software. This description has been added to the Methods (page 19).

We have added Figure 2 —figure supplement 2 to show the locations of all recorded units in AC-PC coordinates (A) and in coordinates normalized to the electrophysiological entry point of STN (B, we followed the method of Sharott et al., 2014). There was a nonsignificant trend toward movement units being located more dorsolaterally in raw AC-PC coordinates (A) but this was not evident when examining the normalized coordinates. We state on page 6 that there was no clear association with unit classification and recording location.

3. The authors use custom use a range of custom designed data analyses to tease out STN neurons with tendencies to respond to differently to parts of the task in the different conditions. In some cases, this is a strength of the study and many of the analyses elegantly pull out features of the data that may not be immediately clear (e.g Figures3 and 4). However, in it's current form it is difficult to establish the effect sizes for other analyses – particularly in the case of the "Turn Units." While the authors go to impressive lengths to show that the differences at the Turn are different, from Figure 3C they seem to be very small and a fraction of those seen in response to kinematic parameters. Currently, many of the novel claims are dependent on this difference (Abstract: "We show that STN activity decreases during action switches, contrary to prevalent theories".). Can the authors provide evidence of the effect size of the responses to task switching as well as being able to say they are significantly different? How many individual units are able to distinguish between specific aspects of the task (such as the example in Figure 2C).4. Given the size of the sample – the authors should provide some evidence that their main findings are scene across the majority of patients. For example, how may units in each cluster are derived from different patients?

While we focused our manuscript on population-level results, these results are indeed evident when examining individual units. Author response table 1 shows the individual unit responses for each population level result. For each unit, we used the cluster mass test to determine if any significant differences were present across the time interval over which population-level changes are seen. For example, Turn Units have decreased firing on Impromptu Turn trials compared to Planned Turn trials between 240 ms before and 80 ms after the turn. The firing rate across this interval contained significant differences between Planned and Impromptu Turn trials on 33% of Turn Units (6/18), and the rest of the units showed a trend in the same direction although they could not reach significance due to limited data for each unit. Our threshold for significance at the individual unit level was p=0.1, as is commonly done (e.g., Mallet et al., 2016). Under this criterion, we would expect 10% of units to show significance under the null hypothesis of no difference between trial types. The effect was seen on 33% of neurons allowing us to confidently reject this null hypothesis (p=0.0012, binomial test). The population level results for firing on medial and lateral trials (we have changed our terminology from ipsilateral and contralateral at the suggestion of Reviewer 3) on Movement Units similarly holds at the individual unit level (and has quite a large effect size, statistically significant in 67% of units). These results are now included in the revised manuscript (pages 7, 9, 11).

**Author response table 1. sa2table1:** 

Population Level Result	Fraction of Individual Units	p-value
Decreased firing rate of Movement Units on Planned Turn trials compared to Impromptu Turn trials, 140 ms after movement onset to 130 ms before turn onset (Figures 2B, 3B)	0.38 (6/16)	5.0 × 10^-4^(binomial test)
Decreased firing rate of Movement Units on Planned Turn trials compared to Reach trials 140 ms after movement onset to 130 ms before turn onset (Figure 2B, 3B)	0.13(2/16)	0.21(binomial test)
Decreased firing rate of Turn Units on Impromptu Turn trials compared to Planned Turn trials 240 ms before to 80 ms after turn onset (Figure 2D, 3C)	0.33(6/18)	0.0012(binomial test)
Decreased firing rate of Movement Units on lateral trials compared to medial trials 10 ms before to 270 ms after movement onset (Figure 5)	0.67(8/12)	1.7 × 10^-7^(binomial test)

While we acknowledge that our sample size of patients was small, the results were consistent across them. The table of unit properties (Supplementary File 2) shows the number of units contributed by each patient and the classification of the units. Movement units were seen in 75% of patients (6/8), and turn units were seen in 75% (6/8), as well. All patients had recordings of Movement and/or Turn Units (i.e., no patients contained zero Movement Units and Turn Units). This is reported on pages 4-5 of the revised manuscript.

5. The analysis of β oscillations/synchrony is potentially very interesting, but I cannot interpret it's relevance without knowing the disease-status of the patients.

All patients had Parkinson’s disease. Their age and pre-operative UPDRS scores are now reported in Supplementary File 1 of the revised manuscript. We now discuss these results in the context of Parkinson’s disease (page 16) , suggesting that our conclusions are more broadly applicable given concordant results between studies of STN in humans with Parkinson’s disease or OCD, and non-pathologic recordings in non-human primates and rodents.

Reviewer #2:The authors report on subthalamic nucleus neural population dynamics and its role in movement planning and execution, especially as it relates to changes in the motor plans (they compared trials of straight reach, planned turn or unexpected turn while recording 39 single units from 8 patients undergoing DBS surgery). They used principal component analysis of firing rates and unsupervised clustering to identify two types of cells – movement and turning units, each contributing specific information regarding movement dynamics encoded in the STN (the third identified cluster was too small to analyze). The manuscript is generally well written and the complex analysis is logically presented although sometimes conclusions are difficult to appreciate from the figures. This type of analysis is increasingly necessary to try and understand the complex dynamics in the nervous system, but the challenge will be to relate to behavior in a meaningful way. The authors should relate their findings to (presumably) parkinsonian pathology especially when discussing synchronization of units with β band oscillations which is the hallmark of the disease.

We very much appreciate the reviewer’s comments and have used them as an opportunity to improve the paper, as we explain below.

– The analysis is presented only for move and turn onsets. Was there a difference in firing rates at Go and Impromptu Turn cues? Also, when was the impromptu turn cue given and did this differ by movement speed (within and across subjects)?

Figure 3 —figure supplement 1 shows average neural activity around the go and turn cues (A and B) compared to the movement and turn onset (C and D) for both unit types. There was no significant difference between trial types around the go cue. However, a clear increase in firing rate is visible beginning about 200 ms before movement onset and about 300 ms after the go cue, suggesting that this response is likely related to movement or movement planning, not the processing of the go cue itself.

The turn cue was presented after the movement onset and before the hand reached the target position, prompting subjects to alter their movement trajectory mid-movement. Turn Units responded similarly in different trial types around the time when the turn cue appeared (B). Movement Units had significantly smaller firing rates on Planned Turn trials than Reach trials between 50 ms before and 170 ms after the expected time of the turn cue (A) because subjects had already begun their turn on Planned Turn trials and movement units track hand kinematics (note that subjects observed the actual turn cue only on the Impromptu Turn trials, but we can compare firing rates around that time across trials).

Because our goal in this study was to determine neural responses in STN to planned and unplanned movements, our key comparisons are with respect to turn onset not the turn cue. Past human studies in STN frequently used go/no-go or stop-signal tasks with button presses or joystick movements as the operant response. Reporting cue-aligned responses is reasonable in those cases. However, our task has the advantage of recording a full, extended movement, which could also be multi-segmented (planned and impromptu turn trials). We can thus parse when exactly movement onset and turn onset occur. Indeed, this is where we also found significant modulation of firing rates.

The turn cue on Impromptu Turn trials was presented when subjects were within a specific rectangular window on their path to the reach target. The center of this window was randomly chosen from a uniform distribution between 35% and 65% of the distance between the fixation and target points. Since the time the turn cue was presented was determined by the subject’s position, faster movements were more likely to result in a sooner presentation of the turn cue. However, as explained in the paper, Turn Units were minimally sensitive to changes in movement in kinematics, including speed and acceleration. The turn cue was presented 206 ± 71 ms after movement onset and 504 ± 154 ms after the go cue (mean ± SD across sessions).

To clarify these points, we have added the figure above to the manuscript (Figure 3 —figure supplement 1) and explain the findings and the conceptual advance on pages 7 and 11.

– "This result was not due to lack of statistical power of the cluster mass test, which showed great sensitivity to systematic trends in our datasets." – It is difficult to assess such statements which occur several times. Could you explain what trends?

We apologize for being vague and have now clarified our approach and results in the revised manuscript. A decoder trained to classify population neural responses performs with ∼75% accuracy (cross-validated) for differences of 0.2 z-scores and performs above chance for differences as low as 0.05 z-scores. These analyses give us confidence that absence of significance is not simply due to low power. We have now included the accuracy with which a decoder classifies population neural responses for individual results in the manuscript.

– "Neuronal spiking in STN is typically phase-locked to β oscillations (Kühn et al., 2005; Weinberger et al., 2006)" – is this because of parkinsonian state? The authors need to acknowledge that subjects had Parkinson's disease (presumably, this is not stated anywhere) so findings may be influenced by pathological parkinsonian state, especially in regards to single cell firing being phase locked to β oscillations which is known to be exaggerated in PD.

Thanks for bringing up this important point. We regret the omission of the fact that all of our patients had medically refractory Parkinson’s disease and were off medication on the day of surgery. While our study took place in these patients that are known to have underlying hypersynchronous β activity (Brittain et al., 2014), β oscillatory activity in STN is not a unique phenomenon in Parkinson’s disease; β activity in the STN has been found in patients with OCD (Rappel et al., 2018). Furthermore, β band activity increases during stopping in both Parkinson’s (Ray et al., 2012) and OCD (Bastin et al., 2014), suggesting that the functional role of β oscillations in action inhibition is not unique to Parkinson’s disease. It is, therefore, notable that we did not find an increase in β induced power or spike-field locking prior to subjects interrupting their planned behaviors. However, our results may still be influenced by a pathological Parkinsonsian state consisting of elevated β activity and generalizing them would require recordings in non-Parkinsonian patients or in animal models without underlying disease. We discuss these points on page 16 of the discussion.

– "Interestingly, these sub-populations are partially spatially segregated within STN, with Movement Units located more laterally…" – provide a depiction of unit locations if commenting on this in the Discussion.

We now provide our methodology for locating recorded units on page 19 of the revision, and we have included Figure 2 —figure supplement 2, which depicts unit locations. There was a nonsignificant trend toward movement units being located more dorsolaterally in raw AC-PC coordinates but this was not evident when examining the normalized coordinates (normalization method was adopted from Sharott 2014). While there may be some segregation of unit types, this is not a strong trend and we have removed this claim from our results and discussion.

– "Further, stimulation of glutamatergic cortical inputs to STN (i.e., the hyperdirect pathway) worsens bradykinesia (Gradinaru et al., 2009)" – that paper showed quite the opposite – stimulation of hyperdirect pathway improved bradykinesia. Please address in your conclusions.

Gradinaru et al.’s findings regarding the effects on bradykinesia caused by stimulation of the hyperdirect pathway were complex. Gradinaru et al. explain that stimulation of the hyperdirect pathway from motor cortex to STN improved bradykinesia, but stimulation of glutamatergic inputs in STN resulted in differing effects depending on the frequency of stimulation. We have now corrected our statement in the revised manuscript (page 18)

– "To compare spiking activity around the turn on the Impromptu Turn trials with spiking activity on Reach and Planned Turn trials, we sampled randomly from the distribution of turn cue onsets on Impromptu Turn Trials to define corresponding analysis windows on Reach and Planned Turn trials." – I understand why you had to do this for Reach trials, but why did you not use the actual turn on Planned turn trials as the analysis window?

Thank you for the great question. Our wording here was not clear enough in distinguishing analyses aligned to the turn *cue* and to the turn *movement* itself. The turn cue is not present on Reach or Planned Turn trials so we sampled from the turn cues on Impromptu Turn trials to generate analysis windows for other trial types. For analyses of the turn movement itself, we used the distribution of the times of onset of turn movement in planned turn and impromptu turn trials to define temporally comparable analysis windows for Reach trials on which there was no turn. We have corrected the confusion caused by our wording in the revision (page 20-21).

– Methods: "Subjects were patients undergoing implantation of deep brain stimulation…" Please specific that these were presumably patients with Parkinson's disease, and were they off dopaminergic medications?

All recordings are from a homogeneous population – patients with medically refractory Parkinson’s disease undergoing surgical implantation of DBS electrodes in the STN. They did not take their anti-Parkinsonian medications on the day of surgery. The description of the disease status is now specified in the abstract, introduction, results, discussion, and methods. Patient age, pre-operative UPDRS motor score, side of recording, and number of single and multi-units recorded per patient are now listed in Supplementary File 1. We also confirm that our conclusions do not critically depend on subjects 1, 2, or 7, who contributed relatively more units than others (page 43).

– Authors report "13.5{plus minus}3.3 trials per trial type per subject" – How may trials of each type were done for each unit recorded? How many units were recorded per patient?

The number of units recorded per patient is shown in Supplementary File 1. The number of trials of each type per unit is shown in Supplementary File 2. Please also see our response to the first comment of Reviewer #1, which includes copies of the tables in Supplementary Files 1 and 2.

– Could not play the second movie

Reviewer #3:Conceptually, even if 'impromptu' units are an actual subpopulation of neurons, a decrease in STN activity just before the execution of the unexpected turn seems consistent, not inconsistent, with previous hypotheses on STN as a "brake" or "global stop" nucleus. I think they could clarify their thoughts on this more in their writing.

Many thanks for bringing up this key point, which we now explain more extensively in the revised manuscript. Briefly, theories positing STN as mediating stop processes predict that stopping is accompanied by increased firing rates. Schmidt et al. 2013 is an example of this, where STN spiking immediately follows a stop cue. In our task, movement speed decreases before the turn on both Planned and Impromptu Turn trials, though more so for the latter (Figure 4 —figure supplement 1). If the Turn Units contribute to stopping, we would expect them to have increased firing rates on Impromptu Turn and Planned Turn trials compared to Reach trials and also higher firing rates on Impromptu Turn than Planned Turn trials. Our data indicate the opposite result; Turn Unit firing rates decrease before impromptu turns and increase before planned turns.

Would like to see more specific examples of sorted units and examples that clearly demonstrate the phenomena they are claiming.

Absolutely. In our revision, we now provide Figure 2 —figure supplement 1, which shows 3 example Turn Units and 3 example Movement Units. S and U indicate the subject and unit identity, and N_t_ indicates trial counts. Asterisks above the time axis indicate periods with statistically significant differences between Planned and Impromptu Turn trials. Although the dynamics of responses vary across units, our main findings are apparent in single units. The Movement Units show increased firing beginning around the movement onset and returning to baseline after the feedback. Meanwhile, the Turn Units show decreased firing in the period around the onset of turn movement on Impromptu Turn trials. The units in panels B and D reach statistical significance, and the unit in F shows a trend in the same direction as the majority of the population (please see our answer to comment 3 of reviewer 1 for a count of the individual units with significant statistics).

All group level plots and panels should include the number of unique participants, the number of individual units, and the number of trials. For example, in Figure 2e PC1 appears to just be the PTH from the unit in Figure 2C. This unit appears to overwhelmingly dominate the population variance. Was PCA performed across all subjects/trials/units?

We have now added the information to the figures. Author response table 2 summarizes the subject, unit, and trial counts for in each figure.

**Author response table 2. sa2table2:** 

Figure	Unique subjects	Units	Trials
1	8	N/A	913
2A,C (example unit)	1	1	107
2B,D (example unit)	1	1	45
2E-H	8	39	2550
3A	8	39	2550
3B	6	16	1117
3C	6	18	1177
4 A,C,E,G	6	16	1117
4 B,D,F,G	6	18	1177
5	3	12	970
6	3	19	1501
7 A,C,E,G,I	6	16	1117
7 B,D,F,H,J	6	18	1177
2 Supplement 1 (example units)	4	6	402
2 Supplement 2	6	31	N/A
3 Supplement 1 A,C	6	16	1117
3 Supplement 1 B,D	6	18	1177
4 Supplement 1	8	39	2550
4 Supplement 2 A,C,E,G	6	16	1117
4 Supplement 2 B,D,F,H	6	18	1177
4 Supplement 3	6	16	1117
5 Supplement 1 A,C,E,G,I,K	3	12	970
5 Supplement 2 B,D,F,H,J,L	3	8	537
7 Supplement 1	8	N/A	1318

Regarding the specific example of Figure 2, Figure 2C shows an individual unit while 2E-H show the firing rates of all units projected on the principal components. The first principal component shown in Figure 2E was recapitulated in many individual units, including the example unit shown in 2C, but the PCs were not based on any individual unit or overwhelmingly shaped by a single unit. The example unit does not dominate the variance; it is representative of it. In fact, removing this unit for the calculation of PCs would make little difference in the outcome. PCA was done on PSTHs of correct trials on all units across all subjects. The PCA procedure is described on page 21 of the manuscript.

Given 39 units / 9 STNs = 4 units per STN. How many subjects had multiple recording sites, what was the average number of units per recording site?

We obtained spiking data from 20 recording sites (on average 1.95 units per site). 62.5% of subjects (5/8) had multiple recording sites. This has been added to the Methods (page 21).

Plots should generally be aligned to impromptu turn cue onset. Without this time alignment, it’s unclear whether the observed changes are related to the cue, the prior movement, or the subsequent movement. Even with this alignment there are some questions about prior events spilling into the 'impromptu' turn condition, depending on the latency of the second command and the nature of the kinematic response of the units in a given experiment.

We have added a new supplementary figure (Figure 3 —figure supplement 1) in the revision to show firing rates aligned to the turn cue (panels A-B) and the onset of turn movement (panels C-D) for the two groups of units. When aligned to the turn cue, there is no difference in firing rates of Turn Units on Impromptu and Planned Turn trials (page 11). However, when aligned to the turn movement itself there is a decrease in the firing rate of Turn Units on Impromptu Turn trials compared to Planned Turn trials. This difference precedes the turn movement. Therefore, this finding cannot be related to the turn cue but is instead related to the turn movement.

In addition, in calculating the PSTHs displayed relative to the onset of the turn movement, all spikes before movement onset (including those around the instruction and go cues) and after feedback are removed prior to calculating the firing rate (discussed on page 21). Furthermore, the turn onset occurs 541 ± 263 ms and 731 ± 217 ms after the movement onset on Planned Turn and Impromptu Turn trials, respectively. It is unlikely that the movement onset would cause firing rate responses in STN cells >500 ms later. Indeed, responses to the movement onset peak <200 ms after this event. The comparison of firing rates aligned to the turn cue and the onset of the turn movement, the time course of firing rate differences, and our methodology in calculating the PSTHs boosts confidence that our results are not contaminated by the effects of prior events.

Because our goal in this study was to determine neural responses in STN to planned and unplanned movements, our key comparisons are with respect to turn onset not the turn cue. Past human studies in STN frequently used go/no-go or stop-signal tasks with button presses or joystick movements as the operant response. Reporting cue-aligned responses is reasonable in those cases. However, our task has the advantage of recording a full, extended movement, which could also be multi-segmented (Planned and Impromptu Turn trials). We can thus parse when exactly movement onset and turn onset occur. Indeed, this is where we also found significant modulation of firing rates. We now include and discuss the cue-aligned results for Movement Units (page 7) and Turn Units (page 11) in the revised manuscript

Figure 1D: When is the go or the cue signal. Unclear how the kinematic data are arranged in time.

Figure 1F: What is the purpose of clustering trajectories? Is this used in the subsequent analyses?

Figure 1E-1F: Single subject or across all subjects?

Figure 2B/D: How do these trajectories differ? Following feedback, the traces look identical. Were stats performed across all principal components?

Figure 2B/D shows an individual unit classified as a Turn Unit using hierarchical clustering. A key characteristic of this type of unit (the mean of which is displayed in Figure 3C) is a relatively flat response profile except around the turn where Impromptu Turn trials have the lowest firing rate and Planned Turn trials the highest. Other Turn Units have similar response profiles (Figure 2 —figure supplement 1). Averaged across all Turn Units, the firing rate is significantly smaller on Impromptu Turn trials than Planned Turn trials between 240 ms before to 80 ms after the turn. Indeed, for this example unit, the mean firing rates on Impromptu and Planned turn trials over this interval are significantly different (p=0.019, Wilcoxon signed rank test). The similarity of responses following feedback strengthens our classification. Statistics in Figure 2E-H were performed jointly over all principal components, as described on pages 24-25.

Figure 3C: Turn versus impromptu turn changes are < 0.2 Z-scores – these are not convincing changes.

While the difference in normalized firing rates between Planned and Impromptu Turn trials may seem small to the reviewer, it is a substantial range of the overall modulation of firing rates, which varies from -0.16 to +0.18 z-scores (please note that the z-scoring was done based on the dynamics throughout the whole trial, not aligned to a particular event). Further, the difference is highly significant (p=0.0031, linear mixed effects model that controls for unit and subject effects), and quite sufficient for decoding. In fact, a decoder trained to separate trial types based on population responses achieves an accuracy of 74%. The decoder analysis and clarifications above are now included in the revision (page 11).

Use of terms ipsilateral and contralateral is confusing. I think they mean medial/lateral or flexor/extensor as these tasks appear to be occurring in the same hand.

Ipsilateral and contralateral trials and responses correspond to medial and lateral, respectively, since patients always used the hand contralateral from which recordings are made. We have now adopted the medial/lateral terminology suggested by the reviewer.

Figure 7E-F – description says that turn trials show a decreased spike synchronization, but this is because β disappears. Also it appears that this happens regardless of trial type (not just during turns). They might consider also including high frequency LFP activity for comparison with β.

That’s an excellent question. We do not disagree that part of the effect of decreased spike synchronization to β oscillations is because β disappears. However, while β synchronization disappears during the movement phases of the task, firing rates increase. If STN spiking were driven only by β oscillations we would have expected firing rates to decrease at this point. Because the data go against this prediction, STN neurons appear to switch from entrainment to β to alternative firing patterns during the movement phases of the trial including around the turn onset. We also agree that the β desynchronization happens regardless of trial type, and this in itself is noteworthy as, based on existing theories, we would have expected to see an increase in β before impromptu turns. We have clarified these points on page 15 of the revised manuscript.

We had limited our analysis to the lower frequency bands of the spectrum because of significant contamination of the LFP spectrum at higher frequencies by spiking itself. We have now implemented the spike-removal algorithm of Boroujeni et al. (2020) to attempt to analyze this spectrum. The results are shown in Figure 7. We do not detect any difference with the cluster mass test.

Page 24 – Bootstrap is defined as sampling WITH replacement, please justify/clarify.

We had meant to state “permutation procedure”, as we did sample without replacement. We apologize for the error and have now corrected it in the revision (page 27).

The authors should explicitly state what disease the diagnoses of the patients (assuming Parkinson's disease in all, since these are STN implants).

All patients had medically refractory Parkinson’s disease. Their ages and UPDRS scores are listed Supplementary File 1. We regret the omission and have now stated patients’ underlying disease status in various sections of the manuscript, including the abstract.

Supplemental Video with 3D PCA visualization, why are the plotted dimensions PC1, PC3, and PC4? Where is PC2?

In Video 1, our goal was to show the separation of the trajectories in a clearer fashion than Figure 2E-H. For the 3D plot, it was necessary to omit one dimension. Since PC2 distinguishes the different trial types the least, we chose to omit it from this plot. It was still included in all statistical analyses of principal component data. We now clarify this point in the caption for Video 1 in the revised manuscript (page 43) and will be happy to generate additional videos that show PC2.

[Editors’ note: what follows is the authors’ response to the second round of review.]

Essential revisions:1) Please comment more on what the 'turning' units are actually encoding, and if they could actually be encoding the acceleration and velocity in the vertical axis? Further, do the results still support the statement that 'one sub-population encoding movement kinematics and direction and another encoding unexpected action switches'. Specifically, Figure 4 indicates that both 'Movement Unit' and 'Turn Unit' firing rate can reflect kinematic parameters with different time lags. The authors argued that (Line 276-278): '… the strength of this relationship (for Turn Units) was considerably weaker than for Movement Units (mean R2 = 0.45 compared to 0.61 for Movement Units). Furthermore, movement kinematics did not accurately predict the PSTHs of these units (Figure 4 —figure supplement 2B, D, and F).' However, it is quite subjective to say the R2 values are 'considerably weaker'. Comparing the plots of A, C and E against B, D, and F in Figure 4 —figure supplement 2, can the author's confidently say that movement kinematics can predict Movement Units (shown in plots of A, C and E), but cannot predict Turn Units (shown in plots of B, D and F)? Furthermore, even if the model fitting is a bit worse for Turn Units compare to Movement Units, did the authors used the same model that fits best for movement units? Or to be more fair, did the authors do model comparison for best fitting models for the two types of Units separately?

Thank you for bringing up these important points. The comment raises three key questions, which we answer in the same order asked by the reviewers. First, we explored the possibility of Turn Units encoding vertical acceleration and velocity, using models containing separate predictors for horizontal and vertical movement, as shown in a new figure, Figure 4 – supplement 4. These models did not indicate preferential encoding of vertical movements and performed poorly compared to the models that used the total acceleration and velocity. For example, linear models with separate terms for vertical and horizontal acceleration achieved an out-of-sample R^2^ of 0.089. Equivalent models for velocity were even worse. Therefore, we do not see evidence that Turn Units preferentially encode vertical acceleration or velocity. These analyses are now explained on page 7 of the revised manuscript.

Regarding the reviewers’ second question about the superior encoding of movement kinematics by the Movement Units, the firing rates of these units were significantly better explained by the movement kinematics compared to the Turn Units. The 95% confidence interval for the out-of-sample R^2^ of the kinematic model with total speed and acceleration was [0.45 0.51] for the Movement Units and [0.085 0.18] for the Turn Units. The 95% confidence interval for in-sample R^2^ was [0.57 0.61] for the Movement Units and [0.44 0.49] for the Turn Units. There is no overlap between these confidence intervals, supporting our conclusion. However, please note that our key point in the manuscript is not that Turn Units are void of information about movement kinematics. Rather, we show that:

1. Movement Units represent movement kinematics better while they lack information about unexpected changes of movement plans (unplanned turns).

2. In contrast, Turn Units have distinct firing rates for unplanned and planned turns but weaker representation of movement kinematics.

This relative double dissociation between the Movement and Turn Units is the basis for the names we have chosen for these two groups of neurons. Finally, it is worth clarifying that we arrived at these two groups of neurons not by searching for the neurons that do or don’t represent movement kinematics and unplanned turns. The groups emerged from an unsupervised clustering of the neurons based on the overall dynamics of their PSTHs. We prefer this natural clustering to an engineered search, even though by implementing such a search to identify neurons with specific response properties we could have improved the double dissociation mentioned above. The emergence of the two natural clusters of neurons with different response dynamics and distinct functional properties bolsters the hypothesis that the STN population is inhomogeneous and has the capacity to contribute to complex mental functions. We now clarify the relative double dissociation on pages 4, 11 and 12, and report confidence intervals of the R^2^ of the kinematic model in Figures 4 – supplements 3-4.

Regarding the reviewers’ third question, we selected the kinematic model with the best out-of-sample R^2^ for both Movement Units and Turn Units. Figure 4 – supplement 4 shows the model fits for Turn Units. The out-of-sample R^2^ for Turn Units in the model with total speed and acceleration is 0.13, and the best fitting model, which has total acceleration as its only predictor, has an out-of-sample R^2^ of 0.23. The same two models produce out-of-sample R^2^ of 0.48 and 0.43, respectively, for the Movement Units. We now report 95% confidence intervals on the in-sample and out-of-sample R^2^ of all competing models in Figure 4 – supplements 3-4. The differential ability of the two groups of neurons for encoding kinematics is clear. Further, for any of the top kinematic models, the residual firing rates of Turn Units showed a significant difference for planned and unplanned turns, 240 ms before to 80 ms after turn onset (for the total acceleration model, p=0.0031, HolmBonferroni corrected linear mixed effect model; discussed on page 7). Thus, our conclusions are robust to the exact choice of kinematic models. We now clarify this robustness on page 21.

2) The authors mentioned that 'Turn Units decreased their firing rates before turn onset on Impromptu Turn trials but increased them before turn onset on Planned Turn trials.' (Line 289-291). However, we don't see this statement being supported by data/figure. Can the authors provide evidence/data for this? Or did we miss anything?

Thank you for another great question. In retrospect, we see that our phrasing has been confusing as it could have implied that the firing rates on planned and impromptu turn trials are increased and decreased, respectively, compared to some baseline period. We simply meant that firing rates on impromptu turn trials are lower than on planned turn trials 240 ms before to 80 ms after turn onset (p=0.0031 Holm-Bonferroni corrected linear mixed effect model, test type). The firing rate for the equivalent times on simple reach trials is in between the other two trial types, but it is not a baseline for the statistical tests. We now clarify this point in the paper and have rephrased this sentence (page 7) and a similar statement in our Discussion (page 9-10) to avoid confusion.

3) Reviewer 3 from the previous round of review raised that 'Conceptually, even if 'impromptu' units are an actual subpopulation of neurons, a decrease in STNactivity just before the execution of the unexpected turn seems consistent, not inconsistent, with the previous hypotheses on STN as a "brake" or "global stop" nucleus. We think the authors should clarify their thoughts on this more in their writing.' We feel this was not sufficiently addressed.

Existing theories about the role of STN are based on studies in which subjects were instructed to withhold uninitiated action plans. A common observation in those studies has been that STN responses increase when subjects halt their planned (but uninitiated) movements, leading to the hypothesis that increased activity is associated with stopping (explained in the first paragraph of the Introduction and page 16 in our Discussion). In general, the reviewer is correct that it is possible for any change in STN activity, including a decrease in firing, to be responsible for stopping. However, to our best knowledge, this has not been observed experimentally in past studies. Our finding of decreased STN firing before changes in ongoing movements does not preclude STN acting to stop or change movements, but they do point to the incompleteness of existing models. We now clarify this point to our discussion (page 10).

4) For the statistical analysis Equation. 1 – Equation. 9, it is not clear whether the authors have used self-programmed scripts for these tests or existing toolbox/functions in Matlab to do this. As far as I understand, Equation. 1 – Equation. 9 are similar to what is implemented in the linear mixed effects models available in Matlab (https://www.mathworks.com/help/stats/linear-mixed-effects-models). This can be applied to each time point, and then using cluster mass test to identify the time window with significant effect. Or alternatively, can the authors specify what is difference between the method used in this study and the linear mixed effects model? The authors mentioned that 'For testing factor ;, the levels of ; are permuted within each unit and without permutation of the other factor. The remainder of the calculation continues as for the one factor case, resulting in separate p-values for each main effect, as in a multi-way ANOVA.' Is it equivalent to repeating the multilevel modeling permuting the levels of m? If the method is based on the framework of multi-level modeling, it might not be necessary to present all the Equations.

Our statistical analyses heavily rely on the math of mixed effects modeling, but we could not have readily used pre-existing packages in Matlab because they do not offer a systematic method to determine the time windows for our analyses. To address this shortcoming, we developed the necessary analyses and their associated code by extending the standard cluster mass test. As a reminder, we sought to determine if different conditions (e.g., trial types and movement laterality) were associated with changes in firing rates, but we did not know *a priori* at which points in time to expect such differences. Furthermore, simple testing of firing rate differences in different windows chosen by examining the data would have resulted in an uncontrolled multiple comparisons problem. Our newly-developed analyses explicitly deal with these multiple comparisons while preserving statistical power. The analysis framework shares many components with the mixed effect models, as astutely observed by the reviewers. Our scripts are available in a public github repository

(https://github.com/dlondon12/InterruptedReachClustMass) and we are working on a methods manuscript that explains the analysis approach in depth.

5) The accurate rate of different conditions was around 0.7. Did the authors only focus on the 'accurate trials' which satisfy the 'accuracy criteria' or considered all trials? Is there any difference between the neural activities for 'accurate trials' vs 'inaccurate trials'?

We did focus all of our analyses on trials which satisfied the accuracy criteria. Those that did not can be split into fixation breaks (which should obviously be excluded as these represent lapses in task engagement) and trials in which the subject performed a movement that did not meet the accuracy criteria. Since our goal was to determine the neural responses to a cue to change movement, we had to ensure that the movement actually changed and in a way that allowed planned and impromptu turns trials to be compared. The median number of inaccurate trials per unit that were not fixation breaks was 19 (aggregated across the three trial types), unfortunately too few to draw conclusions about the neural responses accompanying the inaccurate movements. As expected, adding these trials to our analyses (skipping the exclusion step) does not change our conclusions (e.g., firing rate is significantly smaller on Impromptu Turn trials compared to Planned Turn trials 240 ms before to 80 ms after turn onset). We have now added these points to the Methods section (page 15) and changed the caption for Figure 1B to clarify that it shows the fraction of trials that met the accuracy criteria.

6) What is 'level s' in Equation 9 and Line 748? The ; trial represented trial types, and N represent the different units. What is level s?

Level s refers to the movement direction (target side) of each trial (medial or lateral). This is an orthogonal factor to the trial type (reach, planned turn and impromptu turn trials). The two factor cluster mass test considers the separate effects of each factor. This is now described on page 17 just prior to Equation 4.

7) Can the authors compare the baseline spiking rate of the 'movement related' and 'turning related' units identified by clustering analysis applied on the PSTHs.

The median baseline spiking rate of Movement Units is 35.5 Hz (IQR: 15.5 – 88.0 Hz) compared to 36.6 Hz (IQR: 22.2 – 56.6 Hz) for Turn Units. These rates are not significantly different (p=0.82, Wilcoxon rank sum test). This is now reported on page 4.

8) Figure 2. It is mentioned in the figure legend that 'Shadings in C-H are SE.' The same for Figure 4, 'Shading indicates SE in A-F'. But I didn't see the shadings in these plots? The same for Figure 3 plot B-C, it will be informative to show the SE in the plots.